# Towards variance-conserving reconstructions of climate indices with Gaussian Process Regression in an embedding space

Marlene Klockmann[1], Udo von Toussaint[2], and Eduardo Zorita[1]

[1]Institute for Coastal Systems - Analysis and Modelling, Helmholtz-Zentrum Hereon, Geesthacht, Germany
[2]Max Planck Institute for Plasma Physics, Garching, Germany

**Correspondence:** M. Klockmann (marlene.klockmann@hereon.de)

**Abstract.** We present a new framework for the reconstruction of climate indices based on proxy data such as tree rings. The framework is based on the supervised learning method Gaussian Process Regression (GPR) and aims at preserving the amplitude of past climate variability. It can adequately handle noise-contaminated proxies and variable proxy availability over time. To this end, the GPR is formulated in a modified input space, termed here embedding space. We test the new framework for the reconstruction of the Atlantic Multi-decadal Variability (AMV) in a controlled environment with pseudoproxies derived from coupled climate-model simulations. In this test environment, the GPR outperforms benchmark reconstructions based on multi-linear Principal Component Regression. On AMV-relevant timescales, i.e., multi-decadal, the GPR is able to reconstruct the true amplitude of variability even if the proxies contain a realistic non-climatic noise signal and become sparser back in time. Thus, we conclude that the embedded GPR framework is a highly promising tool for climate-index reconstructions.

## 1 Introduction

Climate indices are important measures to describe the evolution of climate on regional, hemispheric or global scales in a condensed way. They reveal relevant timescales of climate variability and, in some cases, also subspaces that are important for predictability. Paramount examples are the El Niño-Southern Oscillation, the North Atlantic Oscillation and the Atlantic Multi-decadal Variability (AMV). To understand whether the typical timescales and magnitude of climate variability have been stationary over time or whether they have changed, e.g., with anthropogenic climate change, we need a long-term perspective on these climate indices. The index-timeseries must not only cover the historical period of the past 150 years but also the period of interest, e.g., the past 1000 to 2000 years (Common Era). To obtain these long timeseries we need information from so-called climate proxies (e.g. tree rings, sediment cores) in combination with sophisticated statistical models to reconstruct the climate indices from the proxy data. We present a new machine-learning framework for climate-index reconstructions and test its skill for reconstructing the AMV.

The AMV is an important index that describes the North Atlantic climate variability on decadal and longer timescales. Different definitions of the AMV have been developed over time, but the basic definition relies on the low-pass filtered spatial average of sea surface temperature anomalies over the North Atlantic. Observations starting in about 1850 indicate that the AMV varies on typical timescales of 30 to 60 years. The state of the AMV plays a key role for many relevant climate phenomena

such as Arctic sea-ice anomalies (Miles et al., 2014), North American and European summer climate, hurricane seasons and Sahel rainfall (Zhang and Delworth, 2006; Zhang et al., 2007). Both atmospheric as well as oceanic processes have been suggested as possible drivers of the AMV (Clement et al., 2015; Zhang et al., 2019; Yan et al., 2019; Garuba et al., 2018, e.g.). It is not clear, how much of the AMV is generated by internal climate variability and how much is generated by changes in external radiative forcing, i.e., volcanic and anthropogenic aerosols, solar insolation and greenhouse gas concentrations (Haustein et al., 2019; Mann et al., 2021).

The observational period of approximately 150 years is not sufficient to provide a long-term perspective on the AMV or in fact any climate index that describes variability on multi-decadal and longer timescales. Therefore, longer timeseries are needed. These are typically derived from climate reconstructions based on climate proxies such as tree rings, bivalves or coral skeletons (e.g. Gray et al., 2004; Mann et al., 2008; Svendsen et al., 2014; Wang et al., 2017; Singh et al., 2018). This kind of reconstruction is based on statistical models that link the target index with proxy timeseries, using the observational period to calibrate their parameters. The trained models then use the much longer proxy timeseries as input to provide an estimation of the target index in the past.

Existing AMV reconstructions disagree on the amplitude and timing of AMV variability, especially prior to the beginning of the 18th century (Wang et al., 2017). As a consequence, they also provide conflicting views on the AMV response to external forcing (Knudsen et al., 2014; Wang et al., 2017; Zhang et al., 2019; Mann et al., 2022). Possible reasons for this disagreement are numerous. In general, the reconstructed variability will depend on the predictor data, i.e., the number, quality and locations of the proxies. Previous AMV reconstructions differed in their employed proxy networks and types, using only terrestrial or also marine records. As an example, including marine records seems to yield better reconstructions of AMV variability (e.g., Saenger et al., 2009; Mette et al., 2021). Proxy data are only available at a limited number of locations on the globe (see e.g., PAGES2k, 2017), and their availability decreases further back in time. Proxies also contain varying amounts of non-climatic signals, i.e., noise.

Existing reconstruction methods range from very simple linear methods such as Composite Plus Scaling (Jones and Mann, 2004) or Principal Component Analysis (e.g., Gray et al., 2004), over more complex linear methods such as Bayesian Hierarchical Modelling (Barboza et al., 2014) to non-linear methods such Random Forest (Michel et al., 2020), Pairwise Comparison (Hanhijärvi et al., 2013) or Data Assimilation (e.g., Singh et al., 2018). The presence of noise or mutually unrelated variability may result in biased estimations of parameters of the statistical models such as regression coefficients. Especially regression-based methods are known to underestimate the true magnitude of variability, especially on lower frequencies (Zorita et al., 2003; Esper et al., 2005; Von Storch et al., 2004; Christiansen et al., 2009). They also tend to "regress to the mean", i.e. they have difficulties in reconstructing values that lie outside the range of the calibration data. This is further exacerbated by the presence of strong warming trends and shortness of the available calibration period (approximately 150 years).

Thus, robust reconstruction methods are needed in order to produce more reliable estimates of the amplitude of the past variability of the AMV in order to better quantify its response to external forcing. This is also a precondition for an unbiased detection of any 'unusual' observed trends and for the subsequent attribution of those trends to a particular forcing, e.g., anthropogenic greenhouse gases. To this end, we need to design reconstruction methods which are more robust against noise

and, importantly, do not strongly 'regress to the mean' when the predictors become more noisy or scarce back in time. As in many disciplines, machine learning methods have successfully gained traction in the climate reconstruction community (e.g., Michel et al., 2020; Zhang et al., 2022; Wegmann and Jaume-Santero, 2023). Here, we explore the potential of the non-linear supervised learning method Gaussian Process Regression (GPR) for climate index reconstructions. GPR finds growing use in climate applications such as climate model emulators (Mansfield et al., 2020) or reconstructions of sea level fields (Kopp et al., 2016) and global mean surface temperature (Büntgen et al., 2021).

Unlike other machine learning methods, such as neural networks, GPR offers greater transparency and is less of a "black box". The number of free parameters is usually much smaller and ideally the parameters have a more direct physical interpretation. A Gaussian Process (GP) describes a distribution over functions with a given mean and covariance structure. The covariance structure is chosen such that the resulting functions best match a given set of observations. This setup appears as more intuitive and closer to the more familiar family of regression methods than convoluted deep learning structures, which in the end may need additional algorithms for their physical interpretation. GPR's non-parametric nature has the advantage that we do not need to make any assumptions about the (non-)linearity of the underlying reconstruction problem. As a Bayesian method, GPR comes with its own uncertainty estimates, which is a very important feature for paleoclimate applications.

We do not only test GPR as a climate index reconstruction tool but also propose a modified input space for the GPR-based reconstructions. To this end, we embed the entire available dataset (proxy data and the target index) in a virtual space. The location of the data timeseries in this space are based on the similarity between the timeseries. The resulting cloud of data points in this virtual space can be viewed as a temporal sequence of images with missing values. The covariance of the GP describes the cross-correlation between the proxy records and the target index across time and virtual space. We use the GPR to fill the missing values, where we do not have observations of the target index. This approach is somewhat similar to kriging in geostatistics, where two-dimensional fields are reconstructed based on point measurements and a known covariance structure. In our case, the input space is not the geographical space but the virtual embedding space and the covariance structure is learned from the data. This set-up has the additional advantage that it can easily accommodate variable proxy availability in time and that the proxy-related uncertainty can be directly accounted for by the parameters of the GP.

To fully judge the methodological performance and related uncertainties, reconstruction methods need to be tested in so-called pseudoproxy experiments (Smerdon, 2012). Many methods have already been tested in such controlled environments, but the evaluation often lacks a thorough assessment of the method's capability to reconstruct the magnitude of the variability on different timescales. In particular, a reconstruction method must be able to capture extreme phases, again to ascertain whether the AMV is sensitive to sudden changes in the external forcing, e.g., after volcanic eruptions, but also to capture possible large internally generated variations, which could occur independent of external forcing. Here, we test our proposed framework of the embedded GPR in such a pseudoproxy environment and place special emphasis on the method's skill of reconstructing extreme phases and the magnitude of variability of the AMV.

## 2 Methods and Data

### 2.1 Pseudoproxies and simulated AMV index

We generate the pseudoproxies from a simulation of the Common Era (i.e., the past 2000 years) with the Max Planck Institute
Earth System Model (MPI-ESM). The model version corresponds to the MPI-ESM-P LR setup used in the 5th phase of the
Coupled Model Intercomparison Project (CMIP5, Giorgetta et al., 2013). A detailed description of the simulation can be found
in Zhang et al. (2022). The target of the pseudo-reconstructions is the simulated AMV index (AMVI). We define the AMVI
as the spatial mean of annually averaged sea surface temperature anomalies (SST) in the North Atlantic (0 to 70°N and 80°W
to 0°E). The SST anomalies are calculated against the mean over the entire simulation period. We do not further detrend the
AMVI because it is difficult to define a meaningful trend period in the paleo context. In the case of real reconstructions, all
proxies and the AMVI would be available for overlapping periods with different length, and it is not possible to define a
meaningful common trend that could be subtracted from all records.

The pseudoproxies are defined as timeseries of the simulated temperature at the model grid points closest to existing proxy
sites in the PAGES2k database (PAGES2k, 2017). Over land, we use 2m annual mean air temperature, over ocean we use
annual mean sea surface temperature. We do not use all available proxy sites from the PAGES2k data base but only a subset
thereof. We limit our selection of proxy sites to those within the North Atlantic domain (10 to 90°N and 100°W to 30°E) with
annual resolution or finer. Out of these, we further select only those locations at which the pseudoproxies have a correlation
of 0.35 or higher with the AMVI during the last 150 simulation years (in this case, both the AMVI and the pseudoproxies are
detrended before calculating the correlation). The final proxy network consists of 23 pseudoproxies (Fig.1a).

We design three sets of pseudoproxies to account for different sources of uncertainty: In the first test case (TCppp), we
use perfect pseudoproxies, i.e., the pseudoproxies contain only the temperature signal. In the second test case (TCnpp), we
use noisy pseudoproxies, i.e., the pseudoproxies contain additional non-climatic noise. The non-climatic noise is generated by
adding white noise to the perfect pseudoproxies. The amplitude of the white noise is defined such that the correlation between
the noisy and the perfect pseudoproxies is 0.5; i.e., the amplitude of the white noise corresponds to the standard deviation of the
perfect pseudoproxy times $\sqrt{3}$. This is a reasonable choice, as the correlation for real proxies with observations ranges from
0.3 to 0.7. The amount of white noise applied here is also well within the range of other pseudoproxy studies (e.g., Smerdon,
2012). To ensure that the performance with noisy data is independent of the specific noise realisation, we create an ensemble
of 30 noise realisations. In both TCppp and TCnpp we assume that all records are available at every point in time, i.e., that the
network size remains constant in time. In reality, different proxy records cover different periods and the network size is not
constant (Fig.1b). Therefore, we set up a third test case (TCp2k) with realistic temporal proxy availability from the PAGES2k
database and both perfect and noisy pseudoproxies. In all three test cases, the pseudoproxy records have annual resolution. The
reconstruction period corresponds to the last 500 simulation years for TCppp and TCnpp, and to the entire 2000 simulation
years for TCp2k.

To test the sensitivity of the method to the underlying climate-model simulation, we repeat the test cases TCppp and TCnpp
with an analogously derived set of 25 pseudoproxies and AMVI from simulations with the Community Climate System Model

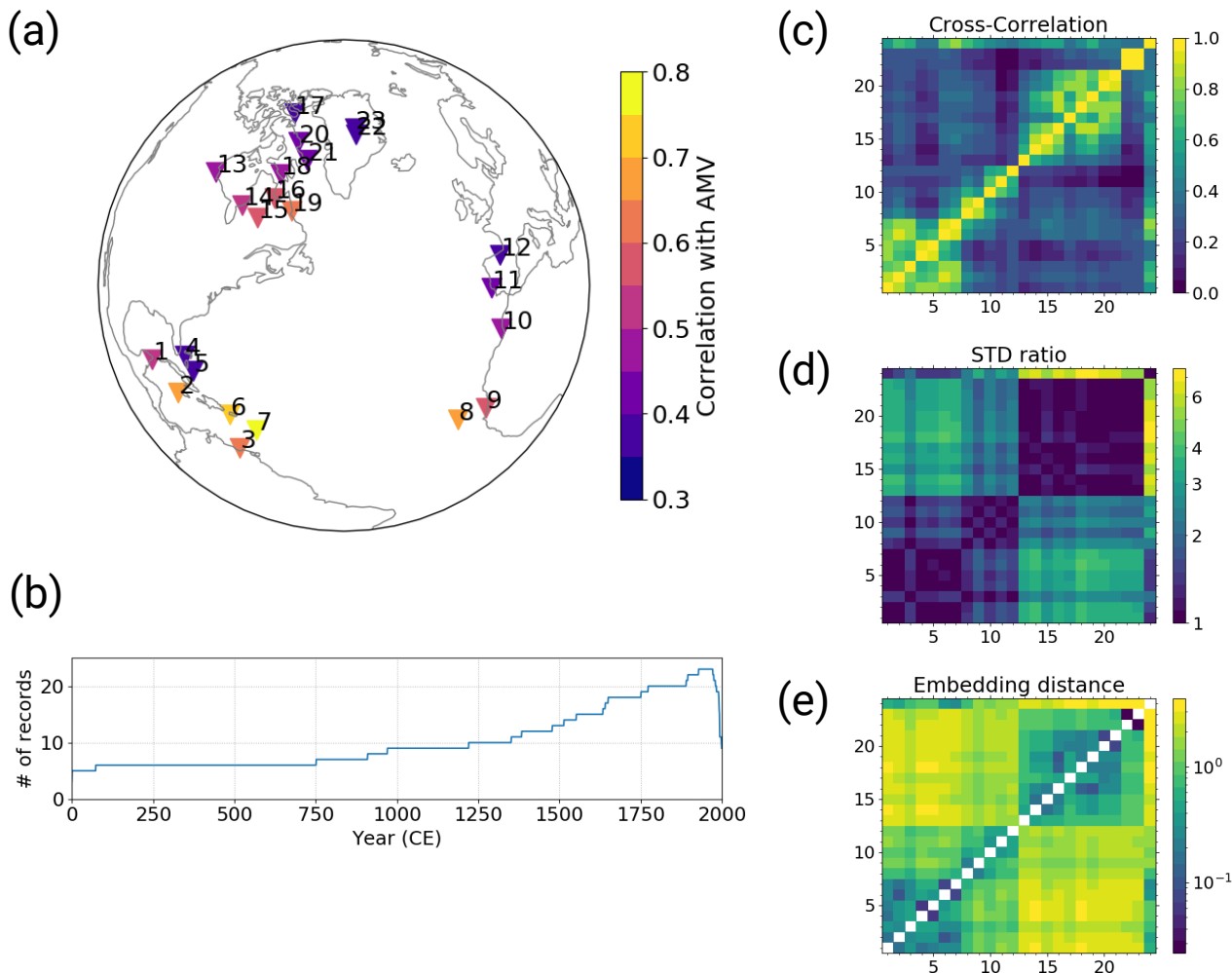

**Figure 1.** The selected pseudoproxy records and resulting distance metrics based on the MPIESM simulation. **(a)** the locations of the records, colour-coded with the correlation between the records and the AMVI during the last 150 simulation years (after detrending); **(b)** the number of available proxy records at the selected locations within the PAGES2k dataset over time; **(c)** cross-correlation; **(d)** standard deviation ratio and **(e)** the resulting embedding distances from the combination of both. Matrix indexes 1 to 23 are the selected pseudo proxy records as labelled in (a), index 24 is the simulated AMVI. The diagonal entries in (e) are left empty because zero cannot be displayed on the logarithmic colour scale.

(CCSM4, Gent et al., 2011). We combine the 'past1000' simulation (Landrum et al., 2013; Otto-Bliesner, 2014) and one 'historical' simulation (Gent et al., 2011; Meehl, 2014) from the CMIP5 suite and use the last 500 years of the combined data set. From the historical simulations, we used the ensemble member r1i1p1. The results are displayed in Appendix B.

## 2.2 Benchmark reconstruction

To have a benchmark for the GPR-based reconstruction in the cases TCppp and TCnpp, we use pseudo-reconstructions with a multi-linear Principal Component Regression (PCR). PCR is well established as a climate-index reconstruction method and has been used e.g., for reconstructions of the global mean surface temperature (PAGES2k, 2019) and the AMVI (Gray et al., 2004; Wang et al., 2017). The selected proxy timeseries are first decomposed into principal components (PCs); the latter are then used as predictors in a linear least-squares regression to obtain the AMVI for those timesteps where proxies and AMVI

overlap. In other words, the AMVI is expressed as a function of PCs of the original proxies (Eq. 1). We do not use all PCs but only retain those with a cumulative explained variance of 99.5%. The trained model can then be used to reconstruct the AMVI for timesteps where we have only proxies available.

$$AMVI(t) = f_{PCR}(PC_1(t), ..., PC_n(t)) \tag{1}$$

## 2.3 Gaussian Process Regression

### 2.3.1 The Concept

Gaussian Process Regression (GPR) is a Bayesian, non-parametric, supervised learning method (Rasmussen and Williams, 2006). Just like a probability distribution describes random variables, a Gaussian Process (GP) describes a distribution over functions with certain properties. A GP is determined by a mean function and a covariance function.

$$f(\mathbf{x}) \sim GP(\mu(\mathbf{x}), k(\mathbf{x}, \mathbf{x'})) \tag{2}$$

The mean function $\mu(\mathbf{x})$ describes the mean of all functions within the GP at location $\mathbf{x}$. In the absence of other knowledge, it is typically assumed that the mean of all functions within the prior GP is zero everywhere. The covariance function $k(\mathbf{x}, \mathbf{x'})$ describes the statistical dependence between the function values at two different points in the input space. The exact covariance structure is prescribed by a kernel function. Kernel functions range from very simple (e.g. linear, radial basis functions) to more complex (e.g., Matern functions, periodic). In principle, there is no limit to the kernel complexity and finding the right

kernel can be considered an art in itself (e.g. Duvenaud et al., 2013). Once a general functional form of the kernel has been chosen (e.g. radial basis function), the specific form is determined by the kernel parameters. Since the underlying GP model itself is non-parametric, kernel parameters are often also referred to as hyperparameters (Rasmussen and Williams, 2006). These hyperparameters are either prescribed a priori if they are known, or learned from the data through optimisation if they are unknown (e.g. through maximum likelihood estimation).

Without being constrained by data, the prior GP is a distribution of all functions with the given mean and covariance (Eq. 2). In order to use the GP for regression and prediction, the prior GP is combined with the additional information from the training data through Bayes theorem (see Appendix A and Rasmussen and Williams (2006) for a more detailed description). Thus, the

posterior GP is obtained, i.e., only those functions are selected that agree with the training data in a given uncertainty range. Predictions at previously unseen input points are then given by that posterior distribution of functions evaluated at those unseen

input points.

### 2.3.2  Finding the right regression space

As described for the PCR, classical climate-index reconstruction methods formulate their underlying statistical model so that the climate index is assumed to be a function of temperature, the proxy values or e.g. principal components thereof. In other words, the regression is performed in temperature/proxy/PC space; the proxies/PCs are the predictors and the climate index

is the predictand. If we reconstruct the AMVI with GPR in this classical setup, the target AMVI becomes the posterior mean function and the covariance is estimated across the proxy space. With the trained GP model, the AMVI can be reconstructed by evaluating the GP at the proxy values that occurred during the reconstruction period. Fig. 2a shows a schematic for the regression in proxy space for an example where the AMVI is given as a function of two pseudoproxy records $p_1$ and $p_2$. In this example, the posterior mean AMVI-function forms a surface in the space spanned by $p_1$ and $p_2$. Note that in our pseudoproxy

experiments we use 23 pseudoproxies (Fig. 1a), so the proxy space is actually 23-dimensional, which is impossible to visualise.

In initial tests, the GPR reconstruction in proxy space did not perform well; the variability of the AMVI was strongly underestimated (not shown). A possible explanation is that GPs are very good interpolators but bad at extrapolating to regions of the proxy space that have not been sampled during training (e.g. upper left and lower right quadrants in the example of Fig. 2a) or where predictors become sparse (lower left quadrant in Fig. 2a). In those cases, the GPR estimation will fall back

to the prior mean function (regression to the mean) and the predictive skill becomes very small. From a mathematical point of view, this setup of the regression in proxy space is also actually not suitable for GPR. Real-world proxies come with large uncertainties, and while GPs are designed to handle uncertain targets, they assume that the inputs are without uncertainty. Therefore, we approach the problem differently and set up the GPR in a way that leverages two GPR strengths: (1) being good interpolators and (2) handling uncertain targets.

In our new approach, we embed the entire available data set (the selected pseudoproxy records and AMVI at all points in time where observations are available) in a virtual space. The cloud of data points can be viewed as a sequence of images in this virtual space. The images contain missing values at timesteps where we do not have AMVI observations available. The climate-index reconstruction problem thus becomes similar to an image-reconstruction problem. The GPR reconstructs the AMVI by filling the missing values based on the surrounding proxy values. In this framework, the GPR inputs are the locations

in the embedding space and the GPR targets are the temperature anomalies of the proxies and the AMVI:

$$\Delta T_i = f_{GPR}(t, \mathbf{x}_i), \tag{3}$$

where $\Delta T_i$ is either a proxy record $p_i$ or the AMVI, and $\mathbf{x}_i$ is the location of the respective record within the embedding space. Fig. 2b shows a schematic view of this embedding space for the example with two pseudoproxies and the AMVI. The location of each timeseries within the embedding space is constant, so that the temporal sequence of data of one particular

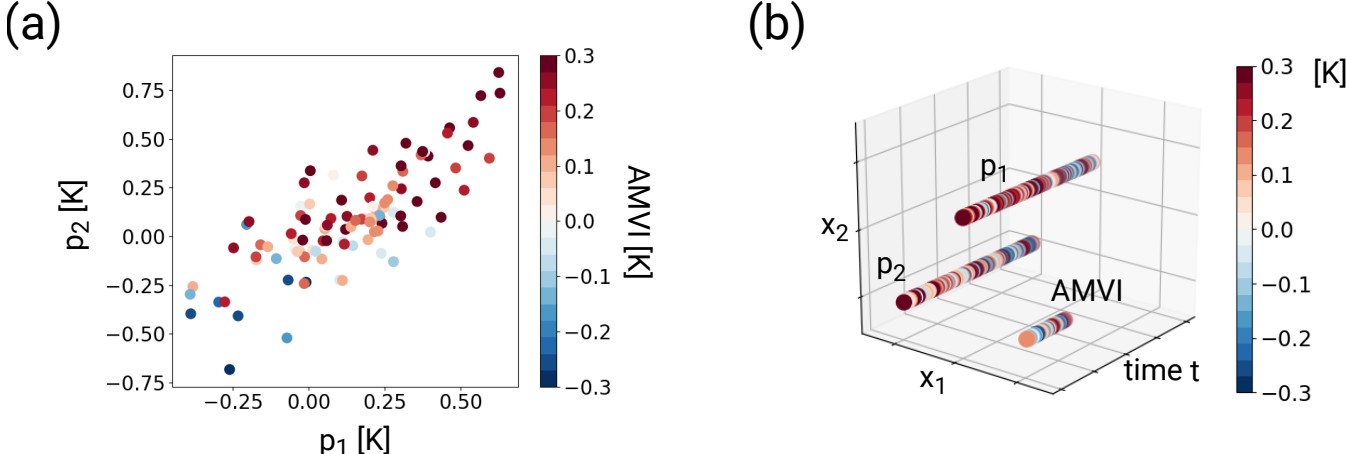

**Figure 2.** Schematic visualisation of the regression spaces for an example with two proxy records $p_1$ and $p_2$ and the AMVI. **(a)** GPR in proxy space: The independent variables are the temperature anomalies of the proxy records; the dependent variable is the AMVI (colour-coded) **(b)** GPR in the embedding space: the independent variables are the locations in the embedding space and time, the dependent variables are the temperature anomalies of the proxy records and the AMVI (colour-coded). In this simplified example, the three timeseries are located such that the distance between them is equal for all respective pairs of records.

timeseries forms a straight line parallel to the time axis. The location of each record is based on its similarity to all other records. The more similar two records are, the closer they are located in the embedding space. To adequately reflect the distances between the proxy records and the AMVI, the embedding space needs to have a dimension of $(q-1)$, where $q$ is the number of timeseries including the AMVI timeseries. This is easiest to understand if one imagines the case where all timeseries have the same distance from each other (as shown in Fig. 2b). To arrange, e.g. three timeseries with equal distances from each

other, one needs a two-dimensional space (spanned by $x_1$ and $x_2$ in Fig. 2b). In case of the MPIESM-based proxy network, the embedding space has thus 23 dimensions (23 proxy records, 1 AMVI). With time as an additional dimension, the resulting space has a total of 24 dimensions. In the following, we will use **r** to refer to a point in space and time, and **x** and $t$ to refer to points in only space and only time, respectively.

We then use the GP to find a function that fits the entire dataset in this virtual space and to interpolate the AMVI at the virtual

locations $\mathbf{x}_{AMV}$ for points in time where we do not have observations. With the right kernel formulation (see Sect. 2.3.4), we can account not only for cross-correlations between the different timeseries, but also for temporal auto-correlation: A data point in the embedding space at time $t_m$ is affected by all other surrounding points in the embedding space at time $t_m$ and to a smaller extent also at times $t_n > t_m$ and $t_k < t_m$. The degree of influence is determined by the distance between the points in the embedding space and the typical length- and timescale of the kernel function. The closer two points are, the larger their

influence.

Thus, the AMVI is still reconstructed based on the information from the pseudoproxies, but we have formulated the problem such that the GP can handle the proxy-related uncertainty correctly, because the pseudoproxies are now targets and no longer inputs. An additional advantage is that we can use this setup with variable proxy availability in time without having to retrain the model each time the proxy availability changes. Instead, the "images" simply have more missing values as the number of proxies decreases further back in time.

### 2.3.3 Defining the distance matrix

Finding the right position **x** for each proxy record and the target index in the embedding space is an important and non-trivial step. Since we care only about the relative distance in the embedding space and not the absolute location, we can specify the distance between each pair of $q$ records (proxies and AMVI) in a distance matrix **D** and determine the coordinates via multi-dimensional scaling (MDS,e.g., Mead, 1992). MDS uses the information of dissimilarity between objects to place these objects in a Cartesian space of a given dimension, such that the distance between the objects in the new space reflects the dissimilarity in an optimal way. In our case, the objects are the proxy records and the AMVI, and the given dimension is 23.

We define the distance matrix based on an appropriate distance metric. This could in principle be any distance metric such as the Euclidian distance or similar. To be used as a distance metric in MDS, a metric must meet the following three criteria: it needs to be (1) positive, (2) zero, when it is applied on the object with itself and (3) symmetric (e.g., Mead, 1992). We chose to define the distance based on the cross-correlation (CC, Fig. 1c) and the standard deviation ratio (SR, Fig. 1d) of the respective records. The SR of two timeseries $p_i$ and $p_j$ is defined as

$$SR_{ij} = \begin{cases} \frac{std(p_i)}{std(p_j)}, \text{ if } std(p_i) > std(p_j) \\ \frac{std(p_j)}{std(p_i)}, \text{ if } std(p_i) < std(p_j), \end{cases} \tag{4}$$

this way, the SR is symmetric, fulfilling the third criterion for the distance metric. Assuming that the records are all positively correlated, the distance measure between two timeseries $p_i$ and $p_j$ is defined as

$$D_{ij} = (1 - CC_{ij}) * SR_{ij}, \tag{5}$$

i.e., the distance will be small when the CC is high and the records have similar amplitudes of variability, and larger when the CC is low and/or the records have very different amplitudes of variability (Fig. 1e). This choice of distance metric outperforms equidistant coordinates and a metric based solely on CC (not shown). With equidistant coordinates all records determine the AMVI to the same degree, regardless of their actual similarity to the AMVI. With a metric based solely on CC, the reconstruction is dominated by records with high variability and the resulting AMVI variability is overestimated. The additional SR-scaling yields improved variability estimates.

The final distance matrix $D$ is then obtained by evaluating Eq. 5 for all pairs of records. For all pairs of pseudoproxies, the distance is estimated from the entire simulation length. For calculating the distance between the AMVI and the pseudoproxies,

we use only the last 150 years (years 1850 to 2000) and linearly detrend both AMVI and pseudoproxies before the calculation. The so determined unitless distances range from 0.02 to 3.91 for the MPIESM-based network (Fig. 1e). The resulting distance matrix is then used as input for the MDS algorithm to obtain the coordinates in a 23-dimensional space. The embedding distance reflects the actual geographical distance to a certain degree. Records that are close in actual space tend to be close also in the embedding space, as they have higher cross-correlations and similar standard deviations (Fig. 1e).

### 2.3.4 Kernel Design and Hyperparameters

We choose a very simple kernel function, the radial basis function (RBF), because we have no prior information that would justify the use of a more complex kernel. Complex kernels would introduce additional uncertainty and reduce the interpretability of the results. We define the kernel as an additive kernel of two RBF components:

$$k1(t_i, t_j) = \sigma_{f,t}^2 \; exp\left(\frac{1}{2}\left(\frac{|t_i - t_j|}{l_{f,t}}\right)^2\right) \tag{6}$$

$$k2(\mathbf{r_i}, \mathbf{r_j}) = \sigma_{f,r}^2 \; exp\left(\frac{1}{2}\left(\frac{|\mathbf{r_i} - \mathbf{r_j}|}{l_{f,r}}\right)^2\right). \tag{7}$$

The final kernel or covariance equation is then given as

$$k = k1 + k2 \tag{8}$$

In Eq. 6 and 7, $|*|$ is the Euclidian distance between two points $t_i$ and $t_j$ or $\mathbf{r_i}$ and $\mathbf{r_j}$. The $l_f$ and $\sigma_f^2$ are the hyperparameters of the respective kernels. $l_f$ denotes a typical lengthscale of the target function, while $\sigma_f^2$ describes the signal variance, e.g., a function with small $l_f$ and large $\sigma_f^2$ will be very wiggly.

The first kernel $k1$ operates on the time dimension only, i.e. it controls how much the neighbouring time steps at one embedding location influence the value at time $t_j$. This could be considered as a mean typical timescale of variability in the dataset. The second kernel $k2$ operates on all dimensions of the embedding space, including the time dimension. This enables interaction between locations at time $t_j$ and neighbouring time steps. This kernel setup outperforms a kernel that consisted only of $k2$ and one where $k2$ did not include the time dimension (not shown). The higher skill of this kernel makes sense if one considers how the kernel design affects the interactions between the different timeseries. Having only $k2$ does not consider that the timescale of auto-correlation may not be the same as the timescale of cross-correlation. It, therefore, makes sense to have $k1$ operate across the time dimension only. If $k2$ operated only across the embedding dimensions, no interaction between different records across time would be possible.

Because $k2$ operates on both the time and the embedding dimensions, we rescale the time steps to be of the same order of magnitude as the distances between the records. This is necessary to allow for the interaction across records and time. Otherwise, the length scale of $k2$ would either be dominated by the time step or by the embedding distance. One rescaled time

step equals the mean of the distance matrix **D**. In the case of the MPIESM-based network (Fig. 1), the mean of the distance matrix is 1.44. For the CCSM-based network (Fig. B1), the mean of the distance matrix is 1.10.

A third additional hyperparameter $\sigma_n^2$ denotes the likelihood or noise variance (see also Appendix A). The noise variance enables the GPR to handle target uncertainty. A small $\sigma_n^2$ indicates that the targets have low uncertainty and the fitted function will be very strongly constrained by the training data. If $\sigma_n^2$ is larger, the targets come with large uncertainty. The fitted function is then less constrained by the training data and more robust against overfitting. Introducing $\sigma_n^2$ is similar to the so-called nugget effect in geostatistics. The noise variance $\sigma_n^2$ is assumed to be the same across all dimensions, i.e., the learned estimate will be the same for all pseudoproxies and the AMVI. This is a simplification, because every pseudoproxy contains its own level of noise. We will show that this simplification is a good first approximation and enables the GPR to handle uncertain pseudoproxies well.

### 2.3.5 GP scaling behaviour

One known drawback of GPs is a bad scaling behaviour of the computing time required to estimate the hyperparameters with respect to the number of available observations, also called batch size. The training time of a GP scales with $n^3$, where $n$ is the batch size. This is mainly due to the necessity to invert the covariance matrix (e.g., Rasmussen and Williams, 2006). Regression problems with more than 1000-10000 observations become difficult to handle with the original GP formulation (hereafter full GP) due to time and computing memory limitations. Even though paleo data sets are not what we would typically call *Big Data*, they can already become challenging for GPs if the reconstruction period spans thousand years or more.

Various GP variants have been proposed to overcome this limitation (e.g., Särkkä, 2013; Hensman et al., 2013). One variant is the so-called *stochastic variational GP* (SVGP, Hensman et al., 2013). The SVGP combines stochastic gradient descent (i.e., training with minibatches), variational inference (i.e., inference through optimisation) and a low-rank approximation of the covariance matrix based on so-called inducing points. Simply put, the inducing points are a small subset of the original dataset that represents the properties of the complete dataset. In other words, the true GP posterior is approximated by a GP that is conditioned on the inducing points. The location of the inducing points in input space can either be prescribed manually (e.g. randomly) or they can be optimised along with the kernel hyperparameters. The training time of the SVGP scales with $m^3$, where $m$ is the number of inducing points (Hensman et al., 2013). Here, we test both the full GP and the SVGP for climate-index reconstruction in the embedding space. In the following, we will refer to the full embedded GP as full emGP and the embedded SVGP as sparse emGP.

### 2.3.6 Technical notes

Our scripts are based on the python package GPflow (Matthews et al., 2017). The hyperparameters are learned through optimisation with the Adam Optimiser, which is a stochastic gradient descent algorithm widely used in machine learning applications (Kingma and Ba, 2014). We use the algorithm as provided by GPflow. For the full GP, we repeat the optimisation step 1000 times. For the sparse GP, we initialise the inducing points as every tenth point in time and the optimise the locations along

with the hyperparameters. We use minibatches with a size of 2000 and repeat the optimisation step 4000 times. The respective number of optimisation steps is sufficient for the estimated likelihood of reaching an equilibrium.

## 2.4 Training and Testing

In the real world, SST measurements are only available since approximately 1850. Therefore, AMV observations are also only available from 1850 to today. We use this criterion to divide our pseudo-data set into training and testing data. The relationship between the pseudoproxies and the simulated AMVI can be inferred only from the last 150 years of the simulation, the remaining years of the simulated AMVI are used for testing. This may not be the most effective way of splitting a data set in the machine-learning context, but it best reflects the actual data availability in the paleo-context.

For the benchmark PCR reconstruction, which takes place in PCR-space, the training inputs are the most recent 150 years of the retained principal components and the training targets are the corresponding 150 years of simulated AMVI (i.e. years 1850 to 2000). In the testing period, the AMVI is reconstructed with the trained regression model and the remaining 350 years of the retained principal components as inputs.

For the emGPR reconstructions, the training inputs are the locations $\mathbf{r}_i$ of the pseudoproxies and the AMVI. For the pseudoproxies, all time steps are used for training (i.e. years 1500 to 2000); for the AMVI only the time steps corresponding to the last 150 simulation years are used for training (i.e. years 1850 to 2000). The training targets are the corresponding values of the 23 proxy records $p_i$ over the full 500 years and the AMVI record over the last 150 years. During training, the kernel hyperparameters and the noise variance are learned. The AMVI is then reconstructed by evaluating the trained emGP at the embedding location of the AMVI $\mathbf{x}_{AMV} = \mathbf{x}_{24}$ and the timesteps corresponding to the remaining 350 simulation years. This approach has the additional advantage that the training set is much bigger than in the classical setup. Thus, we increase the range of climate variability seen during training and reduce the risk of reconstructing climate states that the model has not been trained with.

## 3 Pseudo-reconstructions

### 3.1 TCppp: Perfect pseudoproxies

With perfect pseudoproxies, the best overall reconstruction is achieved by the full emGPR. The reconstructed AMVI closely follows the target AMVI except for the period from approximately 1630 to 1680 (Fig. 3a). This is reflected by the high correlation with the target AMVI (0.93 for the smoothed index). There is a weak negative mean bias, corresponding to 20% of the target standard deviation, which stems mainly from the 50-year period from 1630 to 1680. The GP related uncertainty, as given by the 95th percentile of the posterior distribution, is small for the years 1850 to 2000 where the AMVI has been constrained during training. The uncertainty increases for the reconstruction period. Overall, the posterior uncertainty estimate appears a bit too large - i.e., too conservative - because the true AMVI always lies within the 95% confidence interval. The full emGPR captures the magnitude of variability very well. The standard deviation ratio of 0.93 indicates only a small underestimation

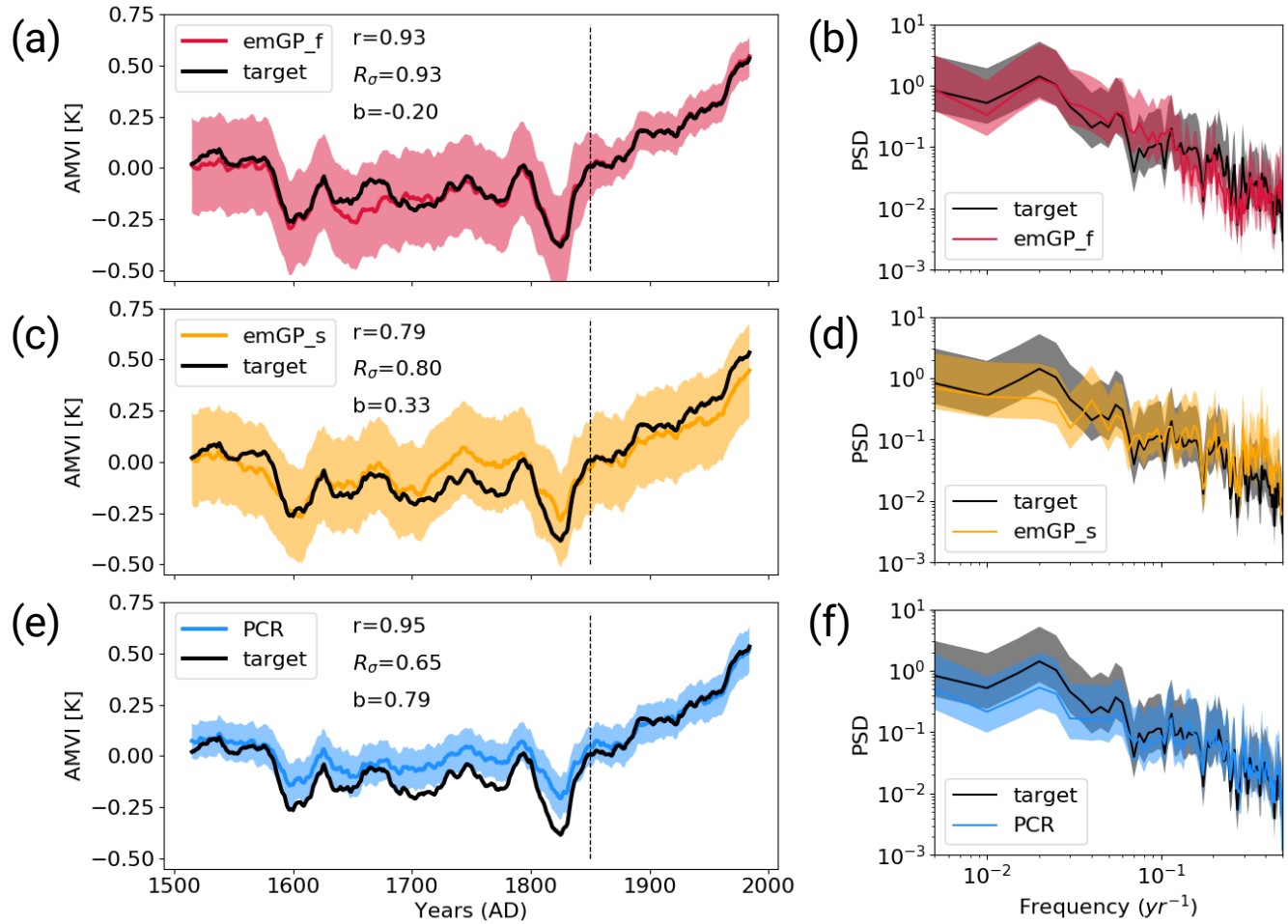

**Figure 3.** Reconstructions with perfect MPIESM-pseudoproxies based on **(a,b)** the full emGPR, **(c,d)** the sparse emGPR and **(e,f)** PCR. Left-hand panels show the smoothed reconstructed and target timeseries. The dashed line marks the separation between training and testing periods. Shading indicates the 95% confidence interval (CI). The CI is determined by the posterior GP distribution for the full and sparse emGP. For the PCR, the CI is derived from the uncertainty in the regression coefficients, which is based on the t-distribution. The metrics $r$, $R_\sigma$ and $b$ denote correlation, the ratio of standard deviations and the bias relative to the target standard deviation, respectively. The metrics are calculated for the smoothed timeseries over the reconstruction period (1500 to 1850). Right-hand panels show the Welch power spectra of the target and reconstructed AMVI. Shading indicates the 95% confidence interval as obtained from the $\chi^2$-distribution. The power spectral density (PSD) is given in $K^2$ yr.

of 7%. Also, the period of very low AMVI following several volcanic eruptions between 1800 and 1850 is well captured, the reconstructed and target AMVI are almost indistinguishable. The spectrum of the reconstructed AMVI agrees well with the spectrum of the target AMVI; the full emGPR captures the variability at all frequencies (Fig. 3b).

The sparse emGPR captures the main features of the target AMVI, but the reconstruction is less accurate (Fig. 3c). The correlation is lower (0.79 for the smoothed index), and there are more periods with larger deviations between the reconstruction and the target. Interestingly, the sparse emGPR has large mismatches during different periods than the full emGPR. The full emGPR has the largest mismatch in the years 1630 to 1680, the sparse emGPR has the largest mismatches in the years 1720 to 1830. The mismatches result in a positive mean bias corresponding to 33% of the target standard deviation. The GP related

uncertainty is the same as the uncertainty from the full emGPR, but in the sparse case, the uncertainty is approximately constant over the entire period. The standard deviation ratio is 0.80, i.e., the variability is underestimated by 20%. This is, e.g. visible for the years 1800 to 1850, where the very low AMVI is not captured as well as by the full emGPR. The spectrum of the reconstruction still agrees well with the target spectrum, but there is a slight overestimation of variability at the very high frequencies and an underestimation at lower frequencies at timescales of 80 to 100 years (Fig. 3d). Differences in the skill

of the full and sparse emGP might partly be explained by different estimates of hyperparameters (see left halves of Fig. 5). We assume that the hyperparameters learned with the full emGPR are closer to the truth. Both the estimated timescale of auto-correlation ($l_{f,t}$) and the signal variance ($\sigma_{f,t}$ and $\sigma_{f,r}$) are underestimated by the sparse emGP.

        The PCR reconstruction achieves the highest correlation (0.95 for the smoothed index) and comes with the smallest uncertainty range, but at the cost of a larger underestimation of variability and a systematic bias towards higher AMVI values,

for periods during which the AMVI is outside of the range of the training period (Fig. 3e). This is especially apparent during the period of the very low AMVI from 1800 to 1850, where the target AMVI lies outside of the PCR uncertainty range. The mean bias corresponds to 79% of the target standard deviation. The standard deviation ratio of 0.65 indicates an underestimation of the variability by 35%. The underestimation occurs systematically at lower frequencies in the multi-decadal range; the high-frequency variability (timescales shorter than 30 years) is well captured (Fig. 3f).

With CCSM4-based pseudo proxies, the results for the full and sparse emGPR are consistent with the MPIESM-based reconstructions (cf. Fig. 3a-d and Fig. B2a-d). The PCR performs much better in the CCSM environment, both the underestimation of variability and the systematic bias to higher AMVI values are smaller in the CCSM4 case (cf. Fig. 3e and Fig. B2e). Also, the low-frequency variability is better captured (Fig. B2f). While the full emGPR clearly outperforms the PCR in the MPIESM case, PCR and the full emGPR perform similarly well in the CCSM4 case. The superior performance of the PCR in the CCSM4

case can be explained by a greater spatial coherence of the underlying CCSM4 temperature field. The leading EOF explains 41% of the total variance in the CCSM4 case and only 27% in the MPIESM case (not shown). The difference in spatial coherence is also reflected in the overall smaller embedding distances in the CCSM4 case (compare Fig. 1 and B1). The fact that the full and sparse emGPR perform about equally well for MPIESM and CCSM4, indicates that the emGPR is more robust to different degrees of spatial coherence in the underlying field. This can be considered an additional strength of the emGPR.

**3.2    TCnpp: Noisy pseudoproxies**

We calculate the reconstruction skill for each of the 30 noisy reconstructions separately and provide the mean and spread of the ensemble statistics. The distribution of the three skill metrics for each reconstruction method can be found in Appendix C.For the full emGP, the individual ensemble members are in reasonably good agreement with the target AMVI (see Fig. 4a

and Fig. C1a). The mean correlation of the smoothed reconstructed AMVI with the target AMVI is $0.89 \pm 0.06$. The mean bias is with $21 \pm 15$ % of the target standard deviation, very similar to the bias from the TCppp case. As with the TCppp testcase, the bias mostly stems from the years 1630 to 1680, where the mismatch between the reconstructions and target is largest. The variability is still captured remarkably well. The mean standard deviation ratio is $1.01 \pm 0.17$, indicating that most ensemble reconstructions contain a realistic amount of variability. The main loss of variability occurs at frequencies higher than decadal(Fig. 4b). The lower-frequency variability range, which is of main interest for studying the AMV, is well reconstructed; the reconstructed spectra lie well within the uncertainty range of the target spectrum.

The sparse emGPR also performs well with noisy pseudoproxies (Fig. 4c,d and Fig. C1b). The overall reconstruction skill is even higher with noisy than with perfect pseudoproxies. This improved performance is also reflected in the estimated timescale of auto-correlation. The estimate for $l_{f,t}$ is now much closer to the estimate from the full emGP (Fig. 5a). We will come back to this improved performance in Sect. 4. The ensemble mean correlation of the smoothed AMVI with the target AMVI is $0.87 \pm 0.09$. Also the mean bias is very small with $11 \pm 9\%$ of the target standard deviation, only a third of the bias from the TCppp case. The mean standard deviation ratio is $0.83 \pm 0.14$, corresponding to an underestimation of the variability by $17 \pm 14\%$. This underestimation is due to a complete loss of power at frequencies higher than decadal (Fig. 4d). But as with the full emGPR, the frequency range of interest for the AMV is well reconstructed.

The PCR still achieves high correlations, but suffers a strong underestimation of variability and an increased systematic bias towards the mean of the AMVI over the training period (Fig. 4e,f and Fig. C1c). These deficiencies of the PCR reconstructions with noisy data have been well documented already (e.g., von Storch et al., 2009). The ensemble mean correlation with the smoothed target AMVI is $0.81 \pm 0.10$. The mean bias of $145 \pm 26\%$ exceeds one standard deviation of the target AMVI. The mean standard deviation ratio is $0.49 \pm 0.09$, corresponding to an underestimation of variability by $51 \pm 9\%$. The loss of variability occurs mainly in the range of frequencies *lower* than decadal, i.e., the frequencies of interest for the AMVI are severely underestimated (Fig. 4f).

The use of noisy pseudoproxies has approximately doubled the width of the 95% confidence intervals for all three methods. The mean uncertainty range over all emGP ensemble members is $\pm 0.57$, which is again too conservative but reasonable given the amount of non-climatic noise in the pseudoproxies. The mean PCR uncertainty range is $\pm 0.21$, which is likely too confident in combination with the large reconstruction bias.

Again, the reconstruction results with the noisy CCSM4-based pseudoproxies are broadly consistent with the MPIESM-based reconstructions (Fig. B3 and Fig. 4). Still, some notable differences occur. The full emGPR has a larger negative mean bias of $75 \pm 17\%$ of the target standard deviation and slightly overestimates the variability on timescales longer than 80 years (Fig. B3a,b). The best reconstruction skill in the noisy CCSM4 case is achieved by the sparse emGPR, with high correlations, a small mean bias and a good estimation of the variability in the decadal to multi-decadal frequency range (Fig. B3c,d). A possible explanation for the higher skill of the sparse emGPR can again be found in the hyperparameters. In the sparse case, the uncertainty from the noisy pseudoproxies was correctly assigned to the noise variance $\sigma_n^2$ (Fig. B4c). In the full case, the large proxy noise was instead interpreted as signal variance ($\sigma_{f,r}$, Fig. B4b). We will return to this point in Sect. 4. With noisy

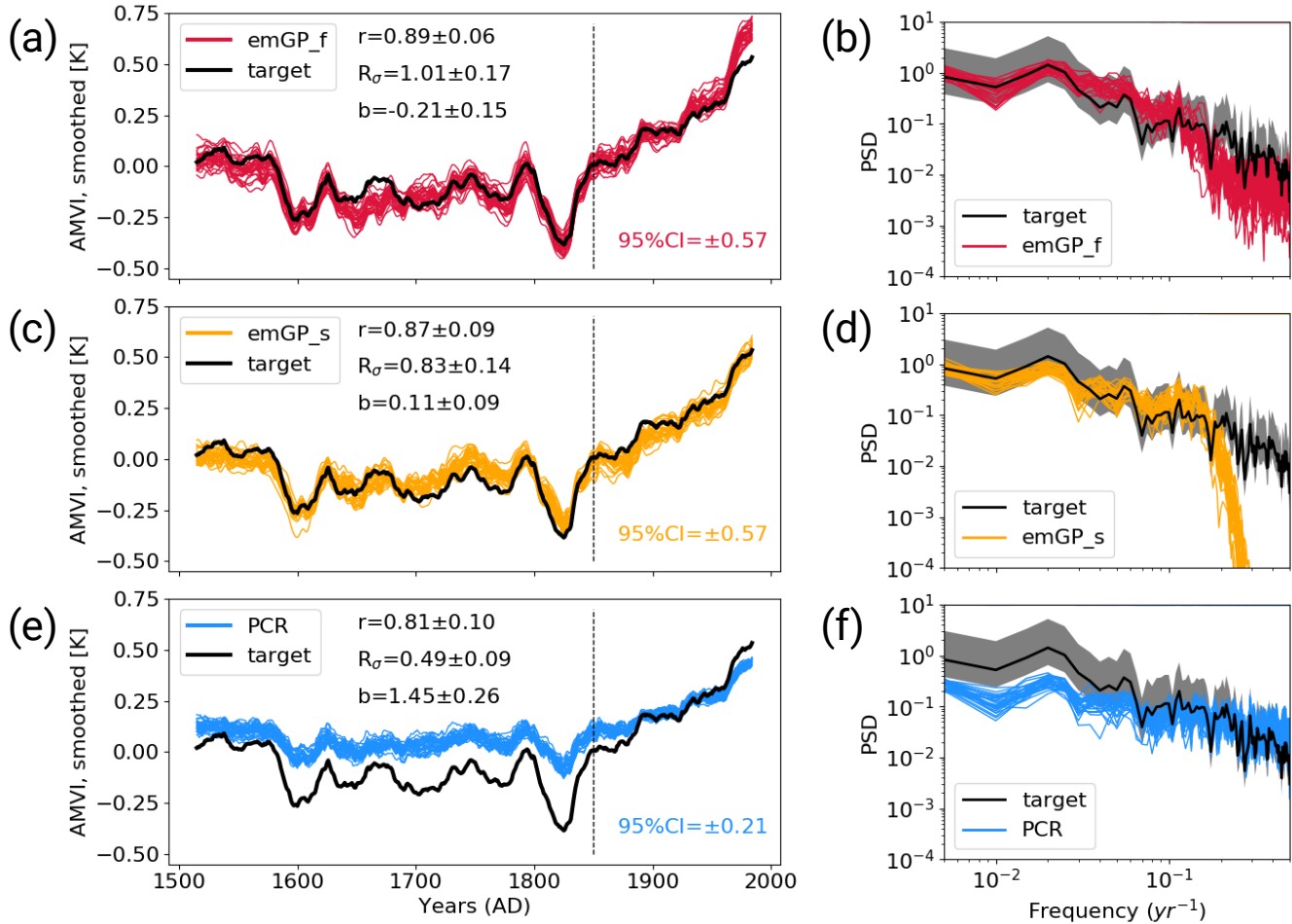

**Figure 4.** Reconstructions with noisy MPIESM-pseudoproxies based on **(a,b)** the full emGPR, **(c,d)** the sparse emGPR and **(e,f)** PCR. Left-hand panels show the smoothed reconstructed and target timeseries. The dashed line marks the separation between training and testing periods. Thin coloured lines show the individual ensemble members. The 95% CI in the lower right of all three panels indicates the CI averaged over time and all ensemble members. The metrics $r$, $R_\sigma$ and $b$ denote correlation, ratio of standard deviations and the bias relative to the target standard deviation, respectively. The metrics are calculated for each smoothed ensemble member over the reconstruction period (1500 to 1850), and the mean and spread ($\pm 2\sigma$) are reported here. Right-hand panels show Welch power spectra of the target and reconstructed AMVI. Thin coloured lines indicate the spectra of the individual ensemble members. Grey shading indicates the 95% confidence interval of the target spectrum as obtained from the $\chi^2$-distribution.. The power spectral density (PSD) is given in K$^2$ yr.

pseudoproxies, the PCR shows the same deficiencies as in the MPIESM case: a strong systematic bias towards the mean of the AMVI during the training period and a strong underestimation of variability on AMV-relevant timescales (Fig. B3e,f).

## 3.3 TCp2k: Realistic PAGES2k proxy availability

Until now, we have assumed that the proxy availability is constant in time. In the following, we assess the reconstruction skill of the two emGPR methods with realistic – i.e., varying – data availability and over the full 2000 years. We do this in three steps: First, we test how the emGPR performs over the full 2000 years with perfect pseudoproxies and constant data availability (i.e., the same as TCppp but over 2000 years). Second, we reconstruct the AMVI with perfect pseudoproxies and realistically varying data availability. To achieve realistic data availability, we clip the annually resolved pseudoproxy records at the start and end years of the corresponding real-world proxy records from the PAGES2k data-base. And third, we test the emGPR with noisy pseudoproxies and varying data availability. The third step, even though still idealised, is closest to representing realistic conditions for proxy-based reconstructions.

Because the full and sparse emGPR differ in the amount of computing memory, we use two different approaches to reconstruct the full 2000 years. In our current computing environment and with the selected MPIESM-based proxy network of 23 locations, the full emGPR cannot handle much more than 500 years at a time. We, therefore, train the full emGPR on the most recent 500 years and use the estimated hyperparameters to reconstruct the AMVI piece-wise in the remaining three blocks of 500 years. For the first step, we actually take the hyperparameters from TCppp (red stars in Fig. 5). For the second step, we estimate the hyperparameters again, to see how much they differ when the data availability changes (red diamonds in Fig. 5). For the full emGPR, they turn out to be very similar, therefore, we use the hyperparameters from TCnpp in the third step in order to save computing time (red dots in Fig. 5). The sparse emGPR can be trained and evaluated over the whole 2000 years at once with reasonable computational effort. Therefore, we retrain the hyperparameters in each of the three steps (yellow diamonds in Fig. 5).

### 3.3.1 Full emGPR

The first step with the full emGPR shows that our approach of piece-wise reconstruction works well. The reconstructed AMVI closely follows the target AMVI also in the years 0 to 1500 (Fig. 6a). This confirms that the hyperparameters estimated from the first 500 years are also representative of the remaining periods (at least in this MPIESM-based setting). The correlation of the smoothed reconstructed AMVI and the target AMVI is with 0.87, a bit lower as in the TCppp case. The mean bias of 8% of the target standard deviation is smaller than in the TCppp case. The variability is well reconstructed, as indicated by both the standard deviation ratio of 1.03 and the power spectrum (Fig. 8a).

With variable data availability in the second step, the full emGPR still achieves a similarly high reconstruction skill (Fig. 6b). The correlation of the smoothed reconstructed AMVI and the target AMVI is 0.88 and the mean bias is negligible. Interestingly, the reduced data availability leads to an overestimation of variability in some periods (e.g., in the years 900 to 1100). This is also indicated by the standard deviation ratio of 1.19. This could be attributed to non-optimal hyperparamters for the reduced

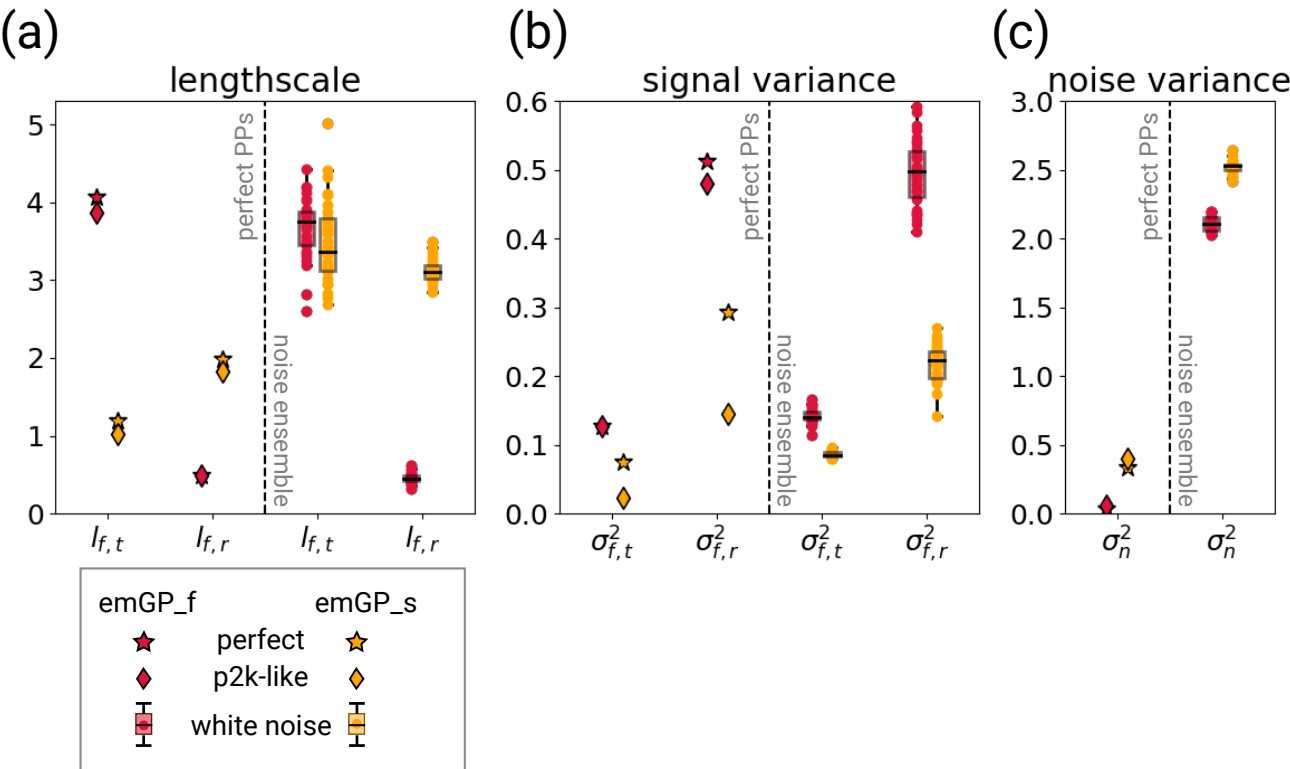

**Figure 5.** The hyperparameters of the two GPR versions for different training periods with perfect MPIESM-pseudoproxies (left halves of the panels) and the different white noise ensemble members (right halves of the panels). The hyperparameters are **(a)** the typical lengthscales $l_{f,t}$ and $l_{f,r}$, **(b)** the signal variance $\sigma^2_{f,t}$ and $\sigma^2_{f,r}$, and **(c)** the noise variance $\sigma^2_n$. The subscript $t$ indicates that the kernel operates only on the time dimension; the subscript $r$ indicates that the kernel operates on all dimensions, including time (see Eq. 6 and 7). The lengthscales are unitless, corresponding to the unitless distance of the embedding space. The lengthscale $l_{f,t}$ can be transformed into years through division by 1.44. The signal and noise variance are given in $K^2$.

proxy availability in this period (see Sect. 4). The power spectrum also shows slightly higher power in the multi-decadal frequency range (Fig. 8b). On the other hand, there is a strong loss of power in the high-frequency range ($> 1/10$ yr$^{-1}$).

The third step confirms that the full emGPR can achieve high reconstruction skill also under realistic conditions (Fig. 6c and Fig. C2a). The mean correlation of the smoothed AMVI reconstruction with the target AMVI is $0.67 \pm 0.07$ and the mean bias amounts to $12 \pm 5\%$ of the target standard deviation. Many periods of extreme high and low AMVI are well captured
(e.g., around year 300 and 1150), but some of these extreme periods are also underestimated (e.g., around year 550). The mean standard deviation ratio is $1.06 \pm 0.15$, indicating an overall realistic level of variability. Especially the variability in the decadal to multi-decadal range is still well captured (Fig. 8c). The only loss of variability occurs again in the high-frequency range on timescales shorter than decadal.

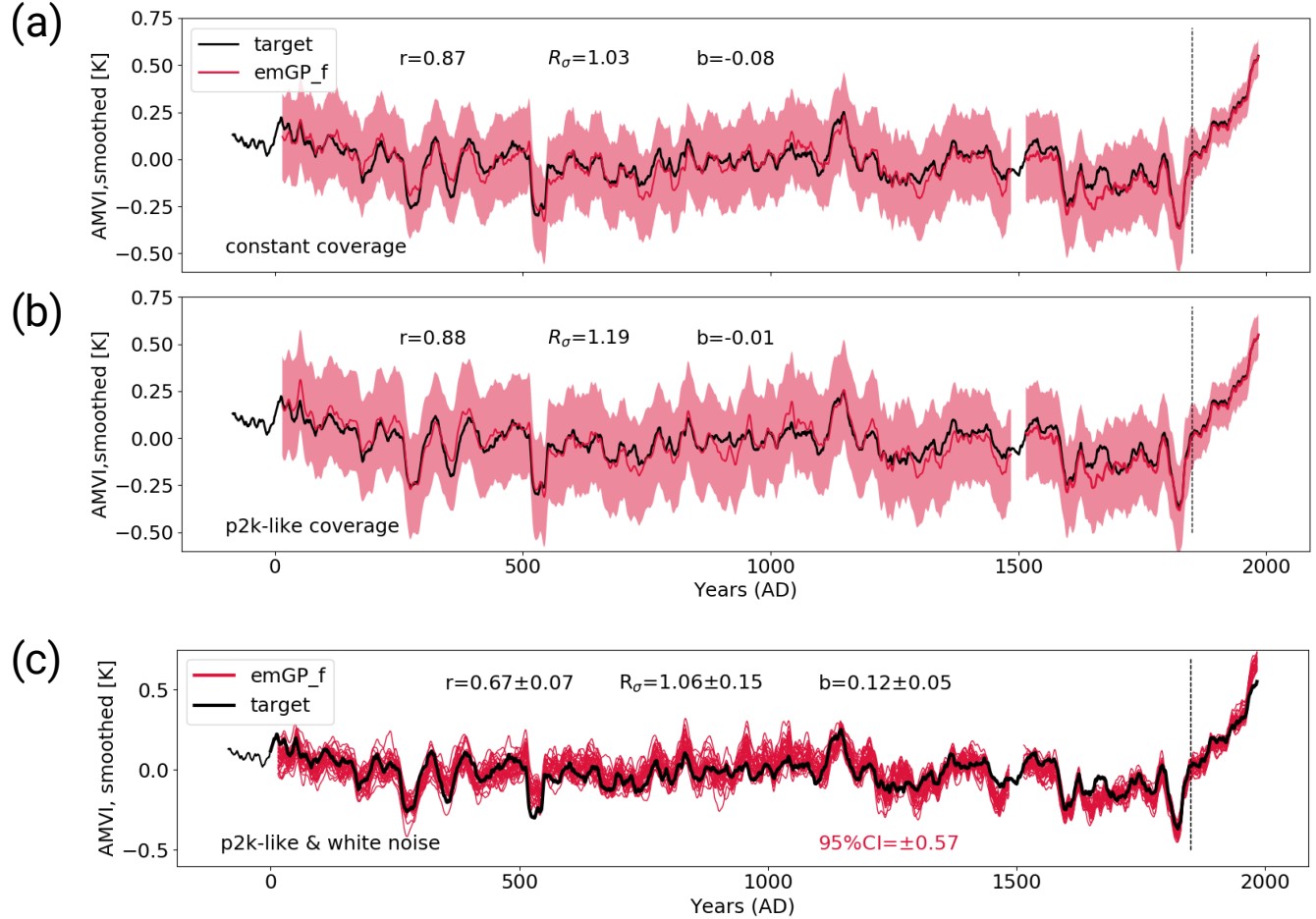

**Figure 6.** The MPIESM-TCp2k reconstructions with the full emGPR. **(a)** first step with perfect pseudoproxies and constant proxy availability. **(b)** second step with perfect pseudoproxies and realistic proxy availability according to the PAGES2k database. **(c)** third step with white-noise added to the proxies and realistic proxy availability.

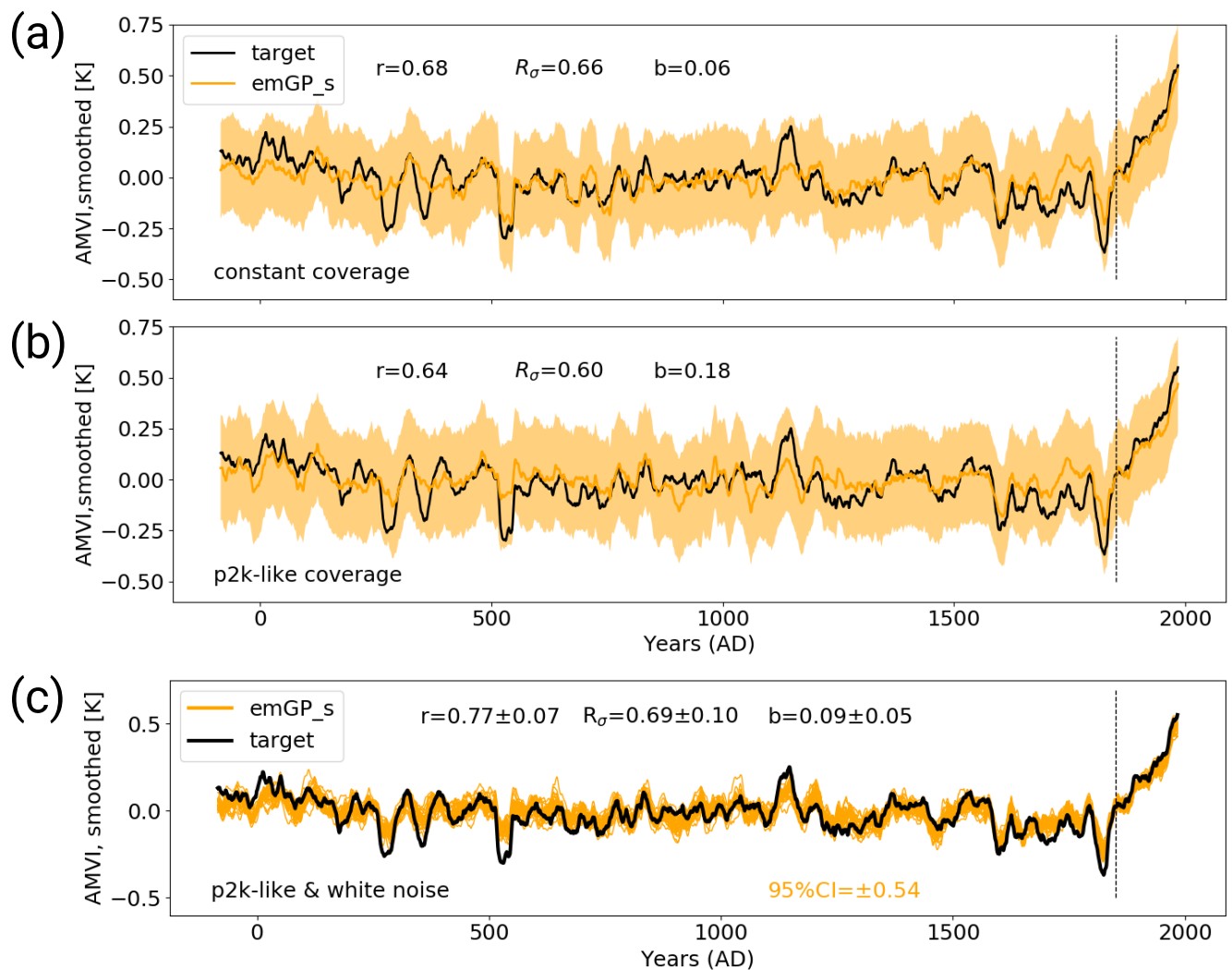

**Figure 7.** The MPIESM-TCp2k reconstructions with the sparse emGPR. **(a)** first step with perfect pseudoproxies and constant proxy availability. **(b)** second step with perfect pseudoproxies and realistic proxy availability according to the PAGES2k database. **(c)** third step with white-noise added to the proxies and realistic proxy availability.

### 3.3.2 Sparse emGPR

In the first step, the sparse emGPR shows a slightly reduced reconstruction skill as compared with the TCppp case. The mean bias is very small, but the correlation is smaller and the underestimation of variability is stronger (Fig. 7a). The variability is underestimated both on interannual and multi-decadal to centennial timescales (Fig. 8,d).

With variable data coverage in the second step, the reconstruction skill of the sparse emGPR remains similar, only the underestimation of variability on multidecadal to centennial timescales increases (Fig. 7b and 8e).

In the third step, the mean correlation of the smoothed AMVI reconstructions with the target AMVI increases to 0.77 $\pm$0.07, confirming again that the sparse emGPR seems to capture some details of the AMVI better in the presence of noise (Fig. 7c and Fig. C2b), and the greater flexibility that comes with a high estimate of noise variance in the GP hyperparameters (Fig. 5c). The underestimation of variability remains large, both on interannual and multi-decadal to centennial timescales (Fig. 8f). Even though the sparse emGPR obtains the worst reconstruction skill in this test case, the overall skill is still higher than that of the

PCR with full data availability in the TCnpp case.

## 4   Discussion

We have tested two versions of GPR in a newly developed input space (embedding space) for climate-index reconstructions in pseudoproxy experiments with increasingly realistic conditions. As a benchmark, we used a PCR-based reconstruction. Under perfect conditions (TCppp), all three methods – full and sparse emGPR and PCR – achieve high reconstruction skill. The

full emGPR outperforms the sparse emGPR and performs at least as well as the PCR. With noise-contaminated pseudoproxies (TCnpp), the full emGPR has the highest reconstruction skill with a realistic estimate of variability on AMV-relevant timescales (i.e., decadal to multi-decadal). The sparse emGPR achieves the second-best reconstruction skill with a realistic mean but increased variance loss. The PCR-based reconstruction is systematically biased to the AMVI values of the training period and suffers a strong loss of variance on AMV-relevant timescales. With realistic proxy availability and noise-contaminated

pseudoproxies (3rd step of TCp2k), the full emGPR is still able to achieve a high reconstruction skill with a realistic mean and variability on AMV-relevant timescales. Below, we re-assess the overall performance of the methods, give possible explanations for differences in reconstruction skill and discuss the wider applicability of the emGP.

### 4.1   Non-climatic noise

Our results indicate that the emGPR (both full and sparse) is able to perform well in the presence of non-climatic noise. This

property can most likely be explained by the hyperparameter of the noise variance $\sigma_n^2$ (Fig. 5c). The noise variance describes the uncertainty in the regression targets. This concept can only be meaningfully applied here because we perform the GPR in the embedding space, where both the proxy timeseries and the AMVI are the regression targets. The noise variance can, therefore, capture the non-climatic signal of the pseudoproxies and give the emGPR the necessary flexibility to filter the non-climatic part of the signal. Comparing the noise variance between the TCppp and TCnpp cases illustrates this quite well. In

the TCppp case, the estimated noise variance (or likelihood) $\sigma_n^2$ lies between 0.1 and 0.4 K$^2$. In the TCnpp case, the estimated $\sigma_n^2$ lies between 2 and 2.7 K$^2$. This does reflect the actual mean magnitude of the noise which we added to the pseudoproxies. The mean variance of the noise of all 23 records is approximately 2.1 K$^2$ (for the individual records, the variance of the added noise ranges from 0.3 K$^2$ to 4.9 K$^2$). Thus, the GPR training procedure seems to be able to learn a realistic magnitude of the added noise.

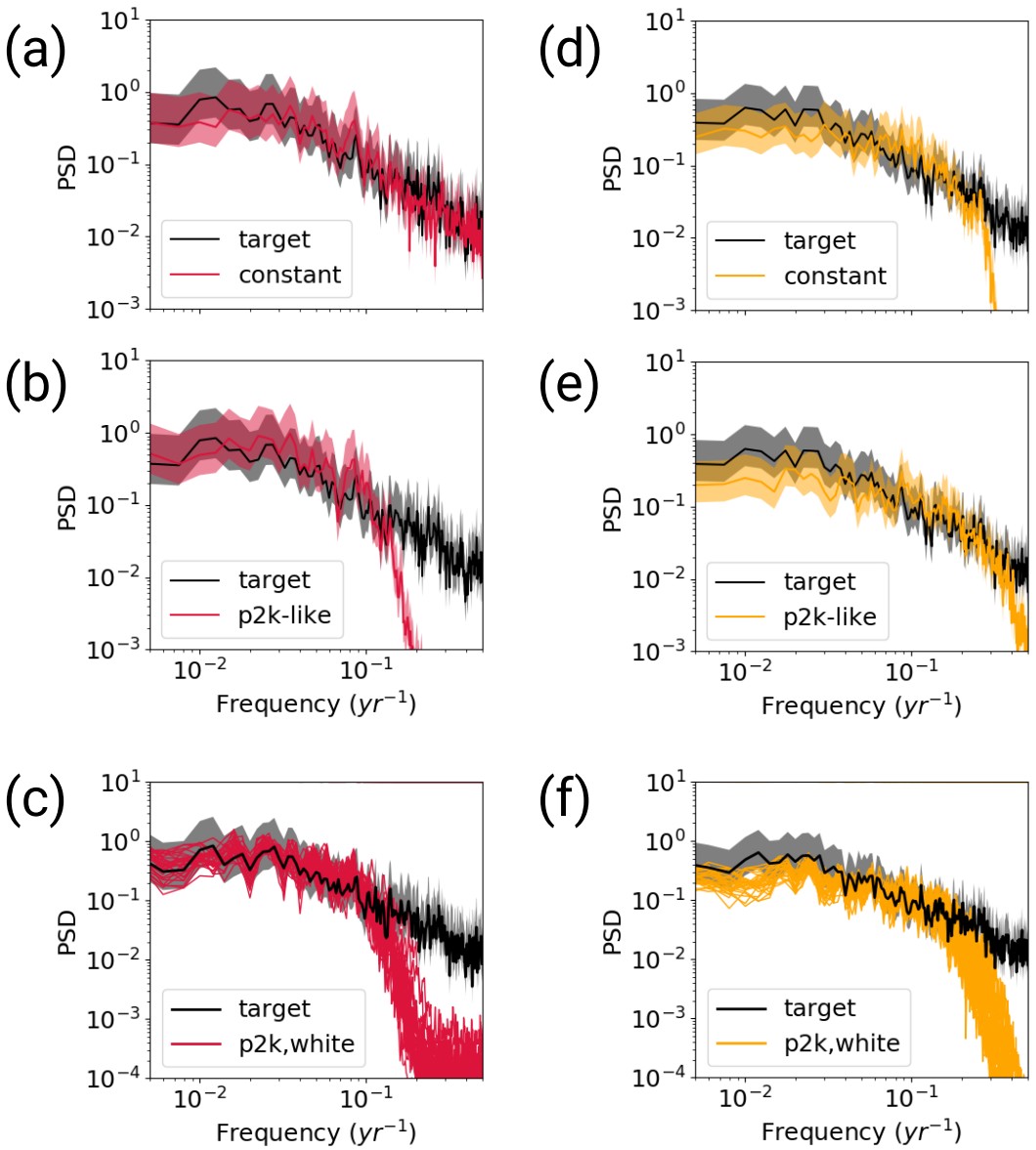

**Figure 8.** Welch power spectra of the MPIESM-TCp2k reconstructions for the full (red) and sparse (yellow) emGPR. **(a,d)** first step with perfect pseudoproxies and constant proxy availability. **(b,e)** second step with perfect pseudoproxies and realistic proxy availability according to the PAGES2k database. **(c,f)** third step with white-noise added to the proxies and realistic proxy availability, grey shading indicates the 95% confidence interval of the target spectrum. The power spectral density (PSD) is given in $K^2$ yr.

Interestingly, the increased flexibility of the GP through the higher $\sigma_n^2$ not only yields robust reconstructions in the presence of uncertain pseudoproxies, it also seems to improve the performance of the sparse emGPR. It is possible that the presence of noise makes the sparse emGP less sensitive to overfitting. To test this, we have repeated TCppp and TCnpp for the sparse emGP with a network consisting of only half the number of pseudoproxies (randomly selected from the original networks). We would expect the difference in skill to decrease with the smaller networks, if overfitting was indeed the reason. The difference

between the reconstruction with perfect and noisy data is indeed reduced with respect to the full networks (Fig. D1). The TCppp reconstruction has a comparable reconstruction skill to single TCnpp reconstruction members with both MPIESM and CCSM4-based pseudoproxies. We, therefore, conclude that the improved performance of the sparse emGP with noisy data can at least partly be attributed to a greater robustness against overfitting in the presence of noise.

     Here, we have only tested white-noise pseudoproxies, i.e. we assume that the noise in the pseudoproxy records is not

correlated in time. The typical noise model for $\sigma_n^2$, which we apply here, also works with the assumption of uncorrelated Gaussian white noise. For real proxies this may not always be the case. There are ways of adapting the noise model to include, e.g., correlated noise (see Rasmussen and Williams, 2006). The embedding space would be a good starting point for this, as we explicitly take the time dimension into account. The noise model would introduce additional hyperparameters and make the calibration more complex. If we simply used our current set-up with correlated noise, the model might interpret some of

the noise correlation incorrectly as actual data-correlation. This could be the subject of a follow-up study.

## 4.2   Hyperparameter estimation and overfitting

The full emGPR achieves generally higher reconstruction skill than the sparse emGPR (one exception is the TCnpp case with CCSM4-based pseudoproxies). This could be expected, since the sparse emGPR approximates the covariance matrix based on only a tenth of the available training data, i.e the subset of selected inducing points. Possibly, the hyperparameters are more

accurately learned from the full dataset. Another possibility is that the location of the inducing points is non-optimal. We have initialised the inducing points as every tenth step in time and then optimised the location during training. We have not tested other setups of the inducing points. It is possible that a higher number or differently selected inducing points would result in a higher reconstruction skill.

     The optimisation of the hyperparameters is an additional source of uncertainty. The learned set of hyperparameters may not

always be the optimal set. We did not make any sensitivity tests regarding e.g. initialisation of the hyperparameters. But the fact that the training of the full emGPR resulted in similar hyperparameters for all three MPIESM-based test cases (TCppp,TCnpp and TCp2k) gives us confidence that the estimated hyperparameters for the full emGPR are accurate. The hyperparameters that have a straightforward physical interpretation, i.e. the typical lengthscale $l_{f,t}$ and variance $\sigma_{f,t}^2$ of the first kernel and the noise variance $\sigma_n^2$, also appear reasonable in their magnitudes in most cases (red stars in Fig. 5). The timescale of the full emGP is on

the order of 2.7 years, which is a reasonable timescale of auto-correlation. As discussed above, $\sigma_n^2$ captures the magnitude of the mean added noise variance across all records. The signal variance $\sigma_{f,t}^2$=0.12 $K^2$ indicates a temporal temperature variability of approximately 0.35 K. For all selected pseudoproxies, the temporal variability ranges from 0.28 to 1.15 K. The estimated $\sigma_{f,t}$ is thus on the lower end of plausible values. The timescale $l_{f,r}$ and variance $\sigma_{f,r}^2$ of the second kernel are less straightforward

to interpret, as they operate across space and time. However, $\sigma_{f,r}^2$ should somehow reflect the mean temperature variability across all records and time. The estimate of $\sigma_{f,r}^2$=0,51 K$^2$ indicates a variability of 0.71 K. This is a fairly close estimate of the actual 0.82 K of the underlying data.

Based on the comparison of the hyperparameters across all experiments, we identify two possible cases of non-optimal hyperparameters: First, the TCppp case with the sparse emGPR, based on both MPIESM and CCSM4 pseudoproxies. Here, the estimated typical timescale $l_{f,t}$ is much shorter than that estimated from the full emGPR (left halves of Fig. 5a and Fig. B4a. And second, the TCnpp case with the full emGPR and CCSM4-based pseudoproxies. In this case, the noise variance was not correctly estimated. Instead, the noise variance was attributed to the signal variance $\sigma_{f,r}^2$. The values of $\sigma_n^2$ and $\sigma_{f,r}^2$ appear switched compared to the other TCnpp experiments (compare right halves of Fig. B4b and c, and Fig. 5b and c). Repeating the CCSM4-based TCnpp experiment with switched $\sigma_n^2$ and $\sigma_{f,r}^2$ slightly increases the reconstruction skill, but the skill remains lower than for the TCnpp with the sparse emGPR (not shown). This illustrates how difficult it is to find an optimal set of hyperparameters.

As with all reconstruction methods, it is possible that the (hyper)parameters learned during training are not completely representative for the reconstruction period. To test for the non-stationarity of hyperparameters in case of changing proxy availability, we have repeated the TCp2k experiment with the full emGP for the years 1000-1500 (Fig.D2). The new hyperparameters indicate a smaller signal variance and shorter auto-correlation timescale (red and grey diamonds in Fig. D3). This improves the reconstruction skill with respect to the original TCp2k experiment. However, non-stationary hyperparameters are difficult to account for with real-world data. By training the emGP with the maximum of available proxy data, we try to get the best mean estimate, but cannot fully avoid the effect of non-stationarity.

### 4.3  Embedding distance

As well as the hyperparameters, also the embedding distances are not constant in time (Fig. D4). Changing cross-correlations may lead to under- or overestimation of individual records during certain periods. Again, this is something which is difficult to account for with real world data, and by calculating the embedding distance over the entire periods of proxy availability we try to find the best mean distance estimate.

Another source of uncertainty is the choice of the distance metric on which the creation of the embedding space is based. We have tested equidistant coordinates, cross-correlation and cross-correlation with standard deviation ratio and selected the latter metric. But of course, other ways of constructing the embedding space could be possible. The optimal embedding space may differ for each proxy network and proxy properties. This is definitely worthy of further investigation.

### 4.4  Climate model dependence

The skill of all three methods, including the benchmark PCR, depends to some degree also on the climate model from which the pseudoproxies are derived. This is a known issue (Smerdon et al., 2011). The full emGPR performs better in the MPIESM-based experiments, the PCR performs better in the CCSM4-based experiments and the sparse emGPR performs about equally well in both model worlds. Source of the different skill could be the differences in the network size and location of the

pseudoproxies and differences in the cross-correlation structure. It is, of course, difficult to say whether a reconstruction with real proxies will behave more like the MPIESM-based experiments or more like the CCSM4-based experiments. But regardless of the differences in skill, the fact that the emGPR has higher reconstruction skill in the more realistic TCnpp case and suffers from a much smaller variability loss than the PCR in both model worlds, gives us confidence that emGPR will also improve the reconstructed variability in a real reconstruction.

### 4.5 Using real proxies and wider applications

The pseudoproxy experiments give a good first impression of how the reconstructions may behave with real proxies. Nonetheless, even though the third step of TCp2k (noise contamination and variable proxy coverage) is already quite realistic, it is still idealised. E.g. in the pseudoproxy setup, we calculate the distance matrix $\mathbf{D}$ based on the whole length of the simulation. With real proxies, each proxy record has a different length and covers a different period. In this case, the distance matrix could be calculated based either on a common time period where all records are available (this could be a very short period), or it could be the period of overlap for each individual pair of records.

In principle, the framework presented here can be applied to any climate index that exhibits significant correlations with local proxy sites. It is thus not limited to the AMVI application presented here. With real proxies, that do not all come in units of °C (e.g., lake sediments, tree ring width, isotope ratios), it might make more sense to standardise all records to unit variance. In this case, the embedding distance would no longer need to include the SR-scaling. A simple dependence on the CC might be sufficient. This remains to be tested.

In order to use this framework for indexes that operate on longer timescales, it might become necessary to include records with lower temporal resolution. This would require subsampling of all records to the lowest common resolution, which is common practice in long-term reconstructions. It might also be possible to train one emGP model for the high-resolution records and one for the low resolution-records. The caveat here is that the observational period is often too short to include enough training data for the low-resolution records. But this is true for all reconstruction methods and not unique to the emGP framework here.

In the TCnpp cases, we created 30 different white noise realisations to estimate the noise-related uncertainty. With real proxies, we, of course, only have one realisation of the data and cannot run noise ensembles. But one could think of other ways of generating ensembles, e.g. with slightly different hyperparameters, slightly different ways of constructing the distance matrix or inclusion of different noise models for $\sigma_n^2$. This would instead give insight into the other more methodological sources of uncertainty.

## 5 Conclusions

We have developed and tested a new method for proxy-based climate-index reconstruction. Our aim was to reduce the underestimation of variability on AMV-relevant timescales (decadal to multi-decadal), which is a common drawback of established reconstruction techniques such as PCR. To this end, we applied Gaussian Process regression and developed a modified in-

put space, which we denoted embedding space. We tested two versions of GPR, a full version and a stochastic variational, i.e. sparse, version. The full version is generally more accurate but comes at high computational costs and can only handle a limited amount of data. As a benchmark comparison, we also computed AMVI reconstructions with PCR.

Under ideal conditions (TCppp: pseudoproxies contain only the climate signal, all records available over the entire reconstruction period), the full embedded GPR performs at least as well as the PCR; in the pseudoproxy experiments based on MPIESM the embedded GPR achieves an even higher reconstruction skill and suffers almost no variance loss. Under more realistic conditions (TCnpp: pseudoproxies contaminated with non-climatic white noise, all records available over the entire reconstruction period), the reconstructions skill of the PCR strongly decreases, and both the full and the sparse embedded GPR clearly outperform the PCR. The GP-based reconstructions have an overall small mean bias and reconstruct the variability on AMV-relevant timescales much more accurately. Under even more realistic conditions (TCp2k: pseudoproxies contaminated with non-climatic white noise, records have different length and cover different periods), the sparse embedded GPR still has an overall small mean bias but suffers a strong variance loss, while the full embedded GPR is still capable of reconstructing the variability on the timescales of interest accurately.

Of course, it remains to be seen how the embedded GPR performs with real proxies. As a next step, we will perform a real AMVI reconstruction based on the PAGES2k proxy network. Based on the results presented in this study, we are confident that climate-index reconstructions can be significantly improved with embedded GPR. A more accurate reconstruction of the mean state and the magnitude of variability will help advance our understanding of AMV dynamics, e.g., especially during periods of extreme cooling following volcanic eruptions.

*Code and data availability.* The extracted pseudoproxy data and the simulated AMVI from the MPIESM and CCSM4 simulations as well as the python scripts for the preparation of the pseudoproxy network, the preparation of the embedding space and the GP regression are provided in the supplement of the paper. The used Python packages Scikit-learn (v.0.19.1), TensorFlow (v.1.12.0) and GPflow (v.1.3.0) are publicly available. The PAGES2k database can be downloaded here: https://doi.org/10.6084/m9.figshare.c.3285353. The CCSM4 past1000 and historical simulations can be obtained from the World Data Center for Climate (doi:10.1594/WDCC/CMIP5.NRS4pk and doi:10.1594/WDCC/CMIP5.NRS4hi, respectively).

*Author contributions.* MK and EZ conceptualised the study. UvT developed the concept of the embedding space. MK implemented the code, performed the pseudo-reconstructions, analysed the results and wrote the first draft of the manuscript. All authors discussed the results and contributed to writing the manuscript.

*Competing interests.* The authors declare no conflict of interests.

*Acknowledgements.* This study has been performed in the context of RedMod (redmod-project.de), a project funded by the Helmholtz Inkubator initiative. The emGPR reconstruction code was run on the gpu nodes of the supercomputer *Mistral* at the German Climate Computing Centre (DKRZ). The MPIESM simulation was run by Sebastian Wagner (Helmholtz-Zentrum Hereon). The analysis was enabled and facilitated by the open-source Python packages Scikit-learn (Pedregosa et al., 2011), TensorFlow (Abadi et al., 2015) and GPflow (Hensman et al., 2013). Figures were generated with Matplotlib (Hunter, 2007).

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

## Appendix A:  Calculating the posterior predictive distribution

Given a set of observed data $\mathbf{y} = y_i = f(\mathbf{x}_i)$, the objective is to provide the probability distribution at a yet unobserved data point $\mathbf{z}$, $f(\mathbf{z})$, conditional on the available observations. This is achieved by the application of the Bayes theorem. Before the application of Bayes theorem, the prior for $f(\mathbf{z})$ is just the assumed probability distribution for the Gaussian process, with mean $\mu_{prior}(\mathbf{z})$ and variance $cov_{prior} = k(\mathbf{z}, \mathbf{z})$. Usually, $\mu_{prior}$ is assumed to be zero without loss of generality (e.g. by taking anomalies from the mean). It is also assumed that observations are a realisation of a *noisy* Gaussian process, which are contaminated by uncertainty in observations, i.e, $y_i = f(\mathbf{x}_i) + \epsilon$. The noise $\epsilon$ is assumed to be Gaussian with variance $\sigma_n^2$ and uncorrelated across the locations $\mathbf{x}_i$. After the application of Bayes theorem, the mean and variance can be calculated according to the following predictive equations (for a detailed derivation see  Rasmussen and Williams, 2006):

$$\mu_{post}(\mathbf{z}) = k(\mathbf{z}, \mathbf{x})^T [k(\mathbf{x}, \mathbf{x}) + \sigma_{\mathbf{n}}^2 \mathbf{I}]^{-1} \mathbf{y} \tag{A1}$$

$$cov_{post}(\mathbf{z}) = k(\mathbf{z}, \mathbf{z}) - k(\mathbf{z}, \mathbf{x})[k(\mathbf{x}, \mathbf{x}) + \sigma_n^2 \mathbf{I}]^{-1} k(\mathbf{x}, \mathbf{z}) \tag{A2}$$

where $\mathbf{I}$ is the identity matrix. These equations can be interpreted as follows. The posterior mean is a linear combination of observations $\mathbf{Y}$ and the process covariances between positions of the available observations and the new position $k(\mathbf{z}, \mathbf{x})$. Usually, the kernel is assumed to decrease with increasing separation between locations. This implies that when the new position $\mathbf{z}$ is out of the range of available observations, the posterior mean will tend towards the prior mean. The posterior variance is smaller than the prior variance, since the available observations reduce the range of likely values of $f(\mathbf{z})$.

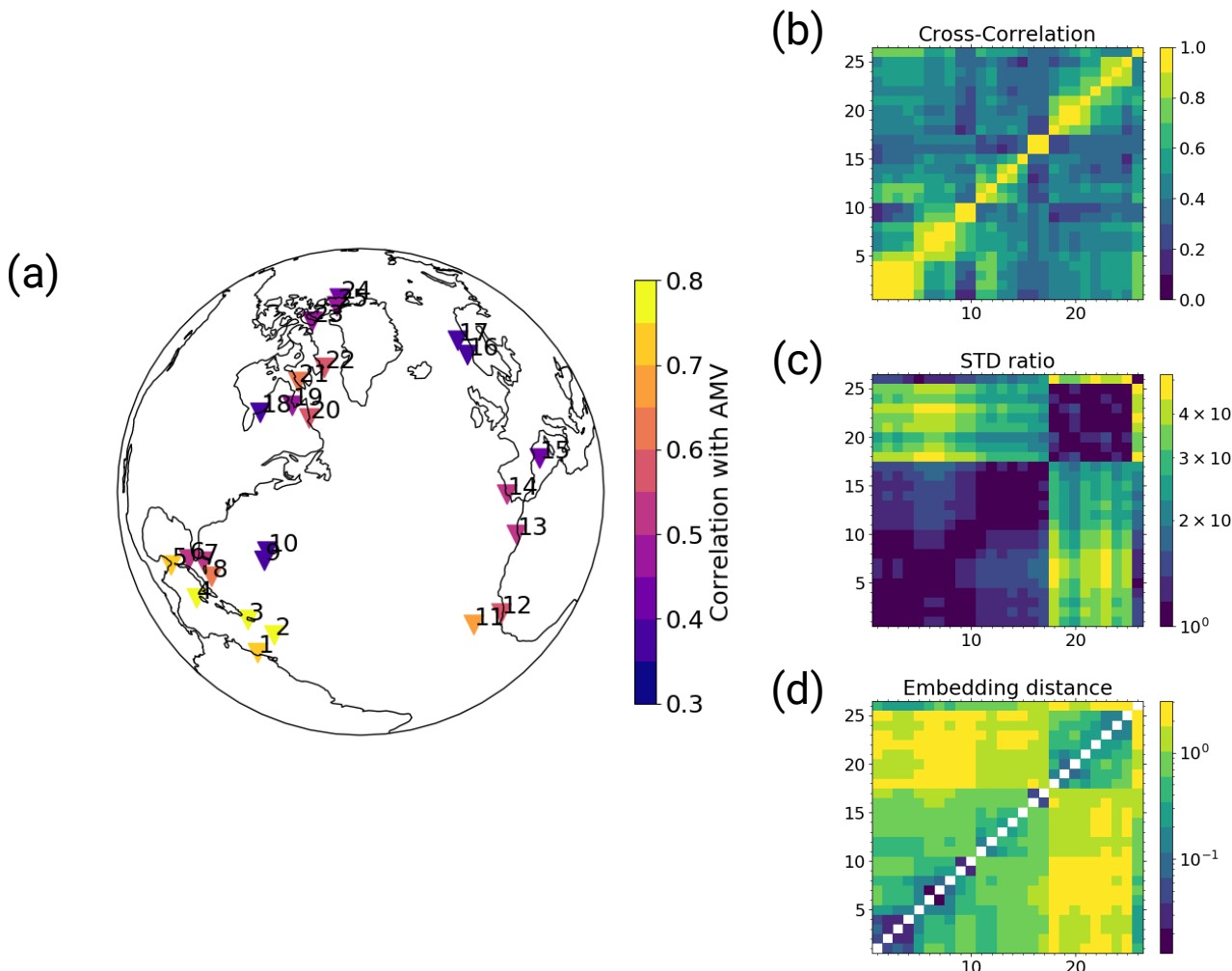

**Figure B1.** The selected pseudoproxy records and resulting distance metrics based on the CCSM4 simulation. **(a)** the locations of the records, colour-coded with the correlation between the records and the AMV during the last 150 simulation years (after detrending); **(b)** cross-correlation; **(c)** standard deviation ratio and **(d)** the resulting embedding distances from the combination of both. Matrix indexes 1 to 25 are the selected pseudo proxy records as labeled in (a), index 26 is the simulated AMV index. The diagonal entries in (d) are left empty because zero cannot be displayed on the logarithmic color scale.

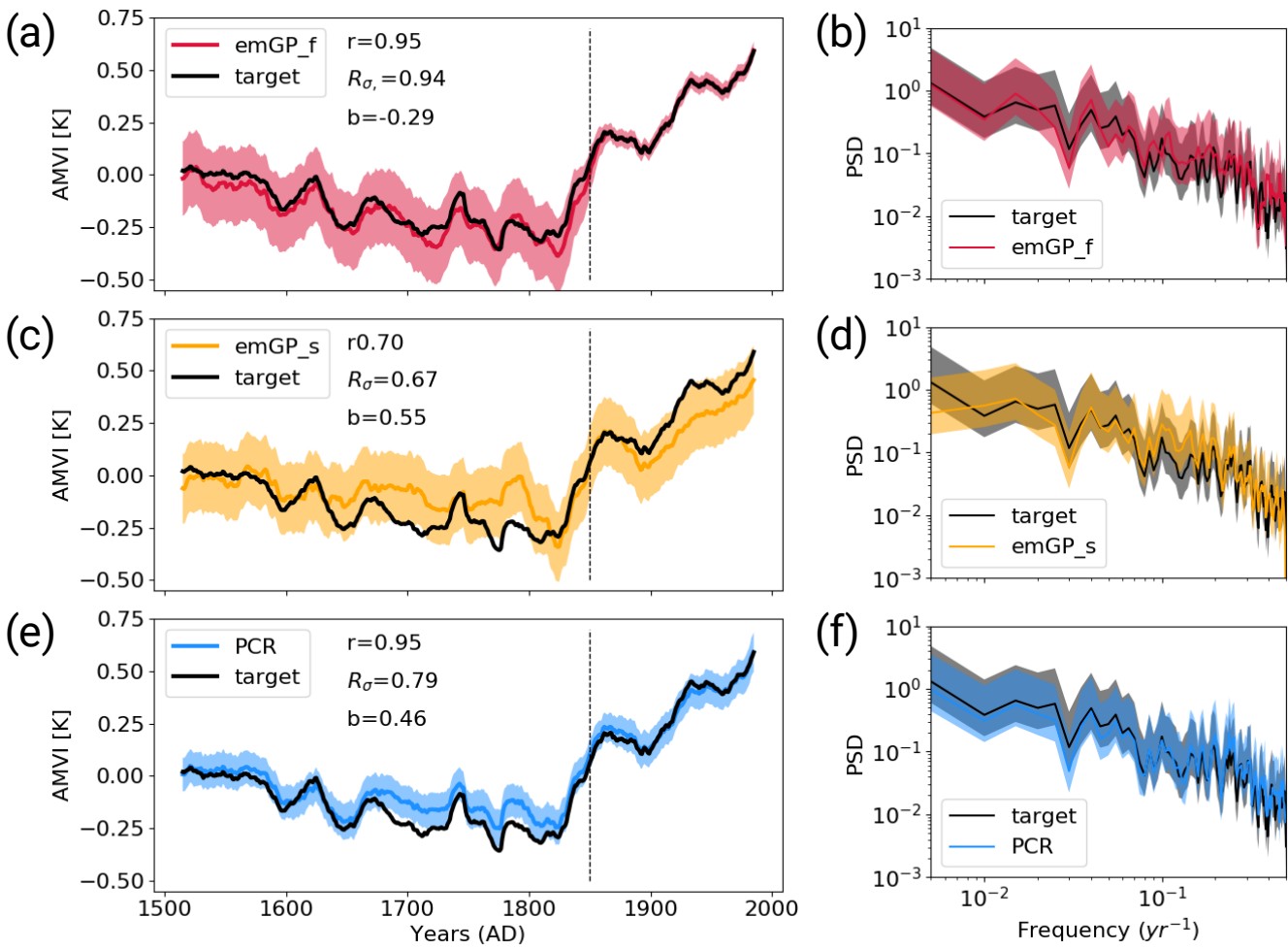

**Figure B2.** Reconstructions with perfect CCSM4-pseudoproxies based on **(a,b)** the full emGPR, **(c,d)** the sparse emGPR **(e,f)** PCR. Left-hand panels show the smoothed reconstructed and target timeseries. The dashed line marks the separation between training and testing periods. Shading indicates the 95% confidence interval. The metrics $r$, $R_\sigma$ and $b$ denote correlation, the ratio of standard deviations and the bias relative to the target standard deviation, respectively. Subscripts $sm$ and $yr$ denote smoothed and unsmoothed data, respectively. The metrics are calculated for the reconstruction period (1500 to 1850). Right-hand panels show the Welch powerspectra of the target and reconstructed AMVI. Shading indicates the 95% confidence interval. The power spectral density (PSD) is given in $K^2$ yr.

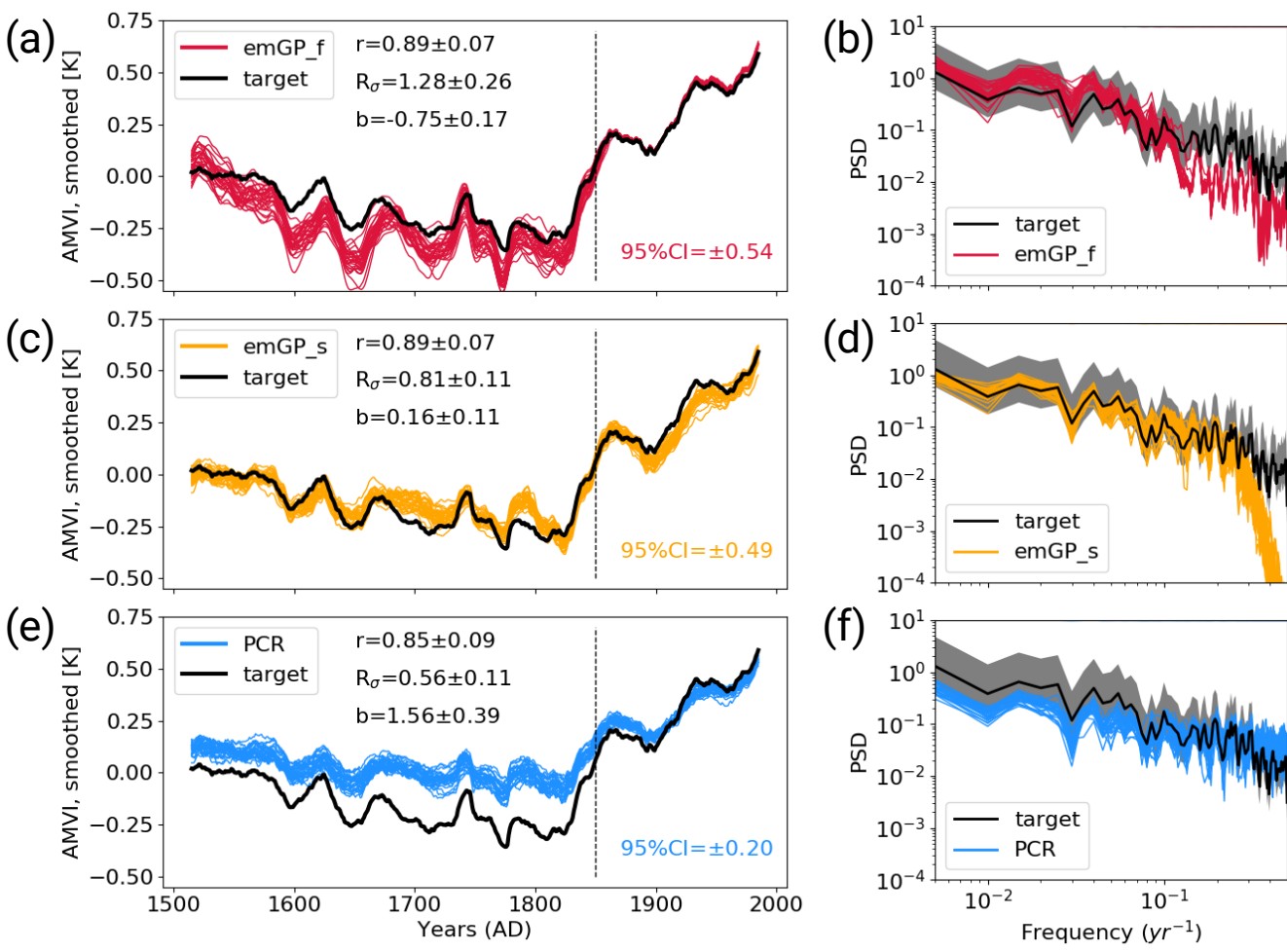

**Figure B3.** Reconstructions with noisy CCSM4-pseudoproxies based on **(a,b)** the full emGPR, **(c,d)** the sparse emGPR **(e,f)** PCR. Left-hand panels show the smoothed reconstructed and target timeseries. The dashed line marks the separation between training and testing periods. Thin lines show the individual ensemble members, the bold line indicates the ensemble mean. The metrics $r$, $R_\sigma$ and $b$ denote correlation, ratio of standard deviations and the bias relative to the target standard deviation, respectively. Subscripts $sm$ and $yr$ denote smoothed and unsmoothed data, respectively. The metrics are calculated for the ensemble mean during the reconstruction period (1500 to 1850). Right-hand panels show Welch powerspectra of the target and reconstructed AMVI. Thin lines indicate the spectra of the individual ensemble members, the bold line indicates the spectrum of the ensemble mean. Shading indicates the 95% confidence interval of the ensemble mean spectrum. The power spectral density (PSD) is given in $K^2$ yr.

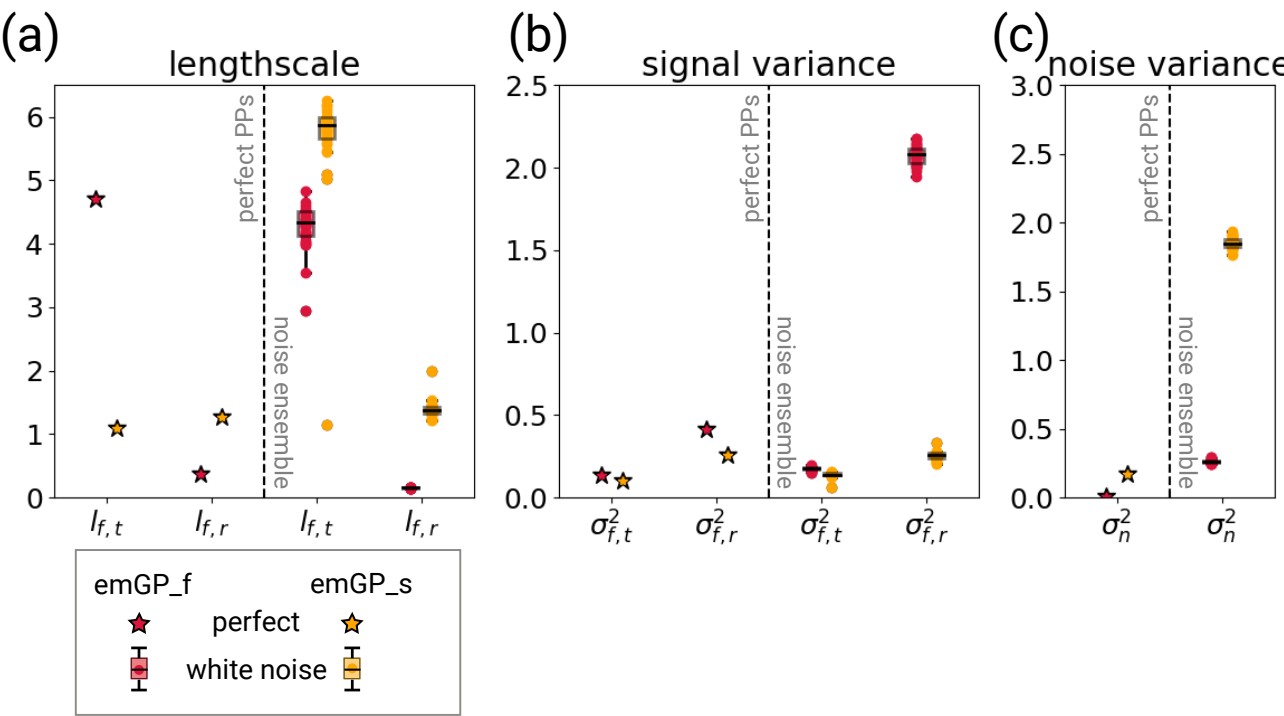

**Figure B4.** The respective hyperparameters for different training periods with CCSM4-based perfect pseudoproxies (left halves of the panels) and the different white noise ensemble members (right halves of the panels). The hyperparameters are **(a)** the typical lengthscales $l_{f,t}$ and $l_{f,r}$, **(b)** the signal variance $\sigma^2_{f,t}$ and $\sigma^2_{f,r}$, and **(c)** the noise variance $\sigma^2_n$. The subscript $t$ indicates that the kernel operates only on the time dimension; the subscript $r$ indicates that the kernel operates on all dimensions, including time (see Eq. 6 and 7). The lengthscales are unitless, corresponding to the unitless distance of the embedding space. The lengthscale $l_{f,t}$ can be transformed into years through division by 1.10. The signal and noise variance are given in $K^2$.

## Appendix C: Metrics of TCnpp ensemble members

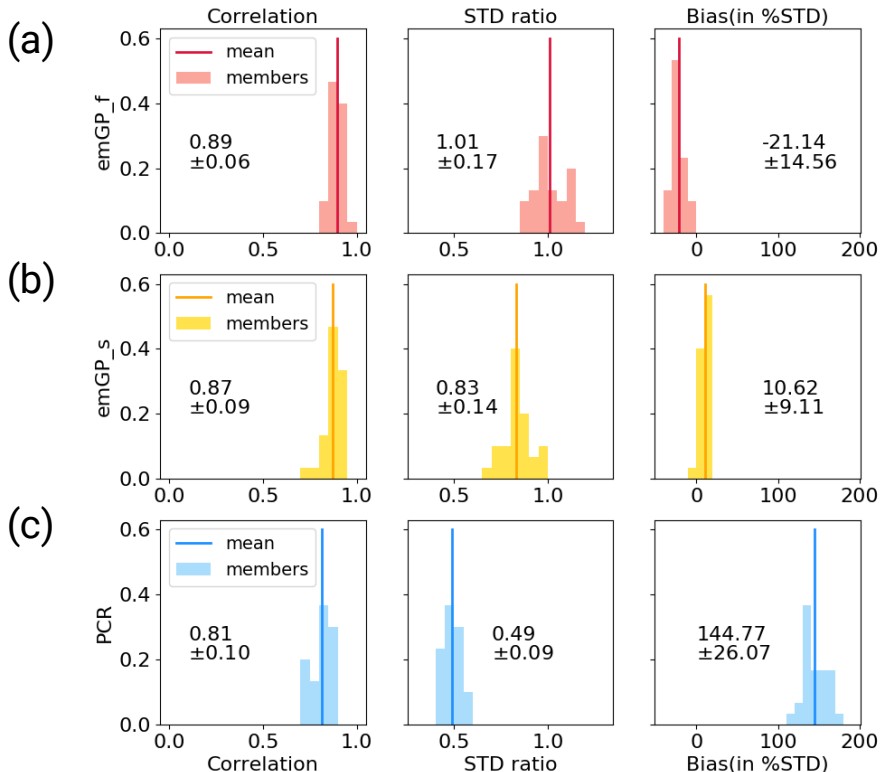

**Figure C1.** Distribution of skill metrics for the ensemble of reconstructions with noisy MPIESM-pseudoproxies with **(a)** the full emGPR, **(b)** the sparse emGPR and **(c)** PCR. The histograms show the respective distributions, the vertical lines indicate the ensemble mean. The printed values denote the mean and spread ($2\sigma$) which are also reported in the text and Fig.4.

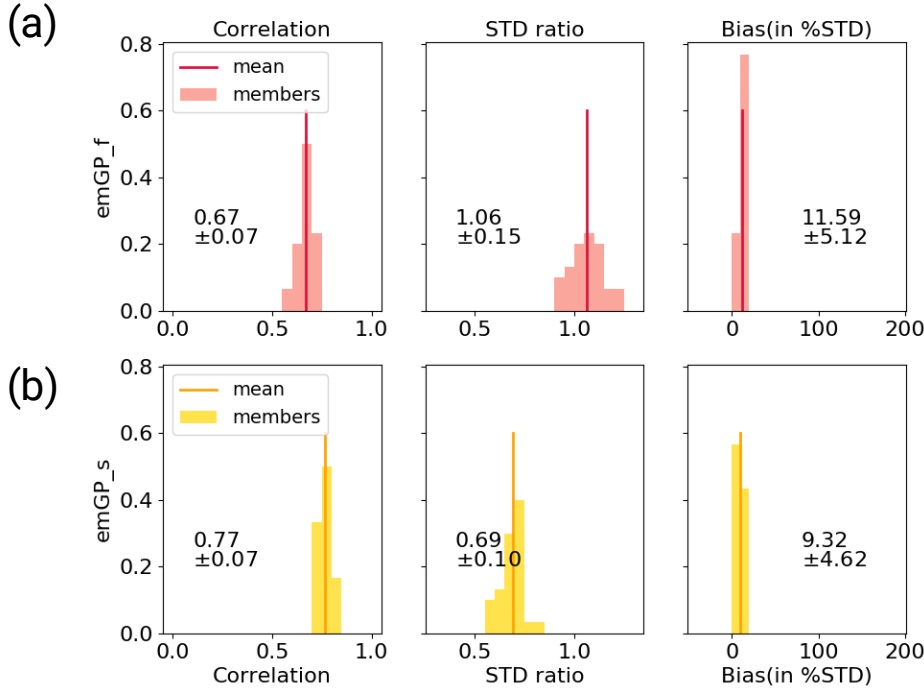

**Figure C2.** Distribution of skill metrics for the ensemble of reconstructions with noisy MPIESM-pseudoproxies and realistic data-availability with **(a)** the full emGPR and **(b)** the sparse emGPR. The histograms show the respective distributions, the vertical lines indicate the ensemble mean. The printed values denote the mean and spread ($2\sigma$) which are also reported in the text, Fig.6 and Fig.7.

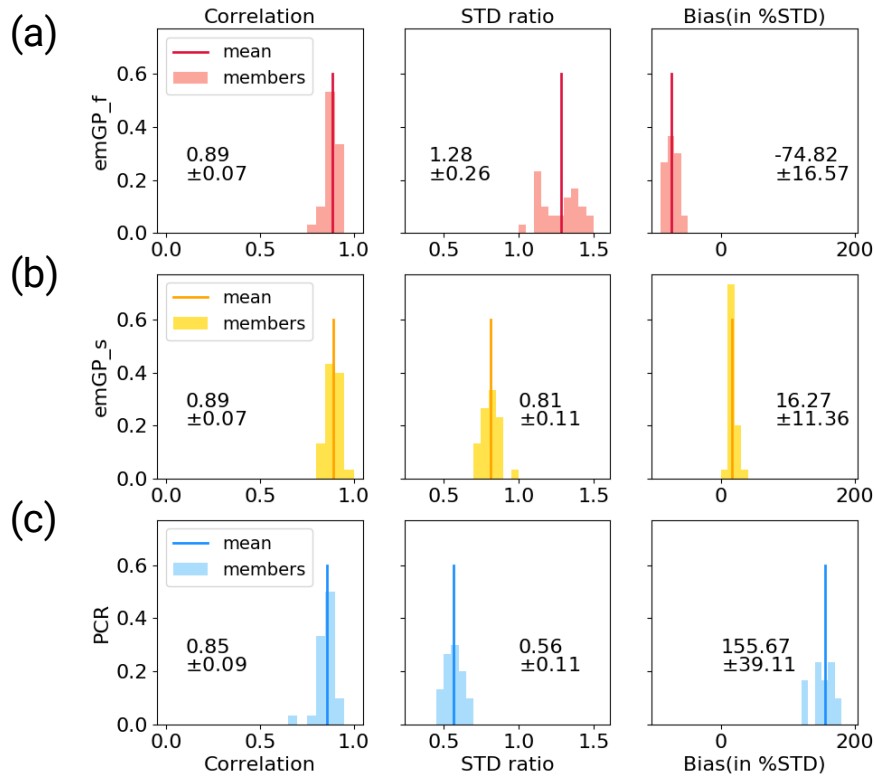

**Figure C3.** Distribution of skill metrics for the ensemble of reconstructions with noisy CCSM4-pseudoproxies with **(a)** the full emGPR, **(b)** the sparse emGPR and **(c)** PCR. The histograms show the respective distributions, the vertical lines indicate the ensemble mean. The printed values denote the mean and spread ($2\sigma$) which are also reported in the text and Fig.B3.

## Appendix D: Sensitivity experiments

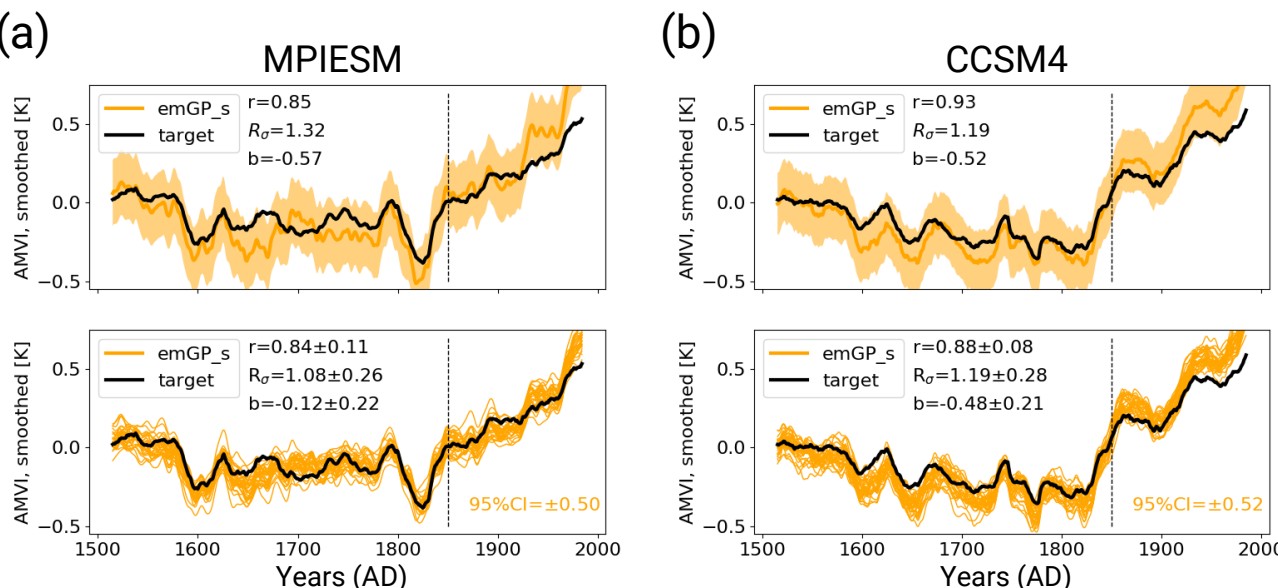

**Figure D1.** Sensitivity experiments similar to TCppp and TCnpp with the sparse emGP but with only half the number of pseudoproxies. Reconstructions based on **(a)** eleven MPIESM-based pseudoproxies and **(b)** twelve CCSM4-based pseudoproxies. The upper panels show the reconstruction with perfect pseudoproxies, the lower panels show the reconstruction with noisy pseudoproxies.

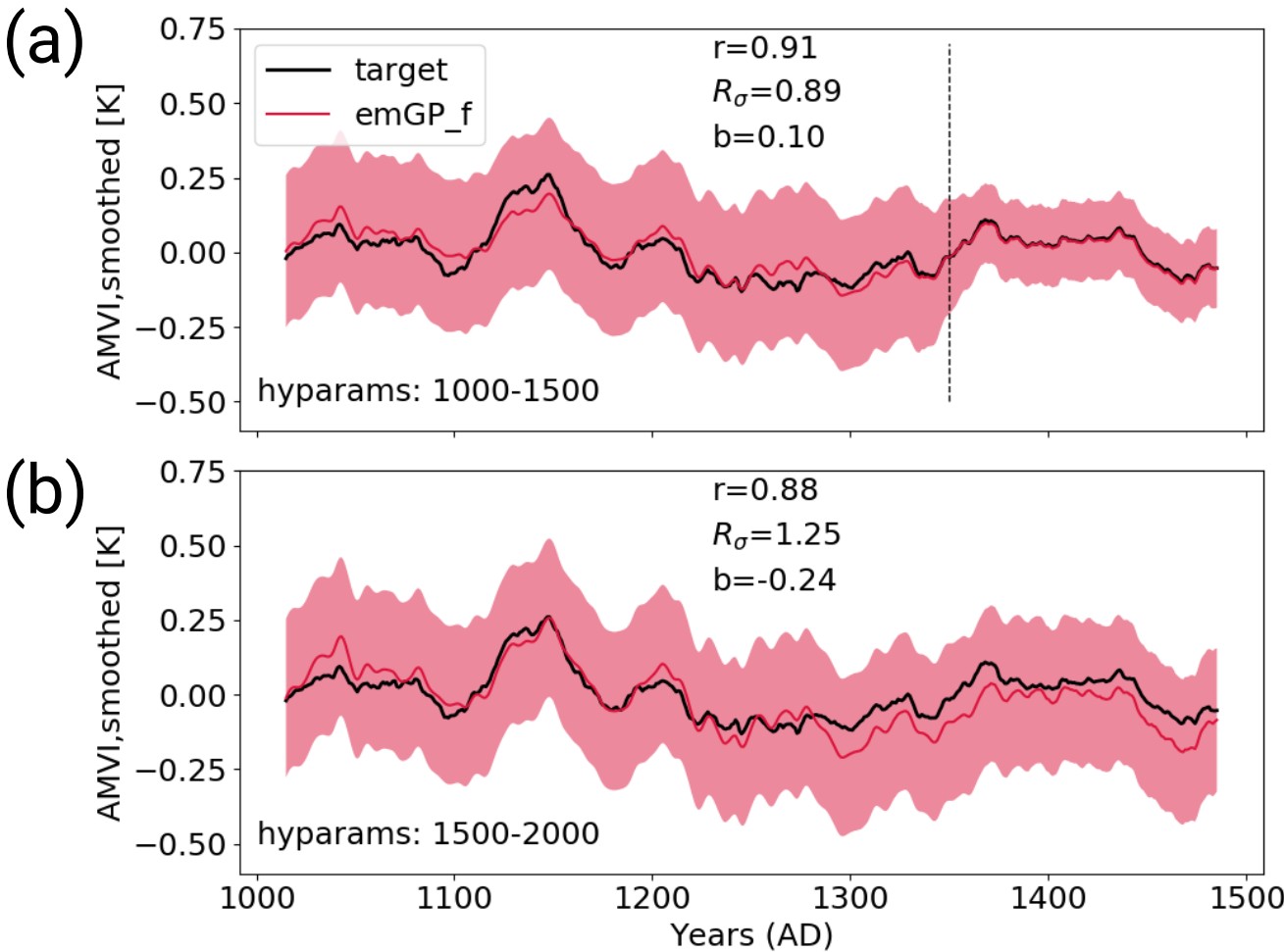

**Figure D2. (a)** Sensitivity experiment similar to TCp2k with the full emGP but the hyperparameters are estimated for the period 1000-1500, where the proxy availability is strongly reduced (see Fig. 1b). The AMVI during the years 1350-1500 has been used for training. **(b)** AMVI reconstruction for the same period from Fig. 6b for comparison. Here, the hyperparameters were estimated for the period 1500-2000.

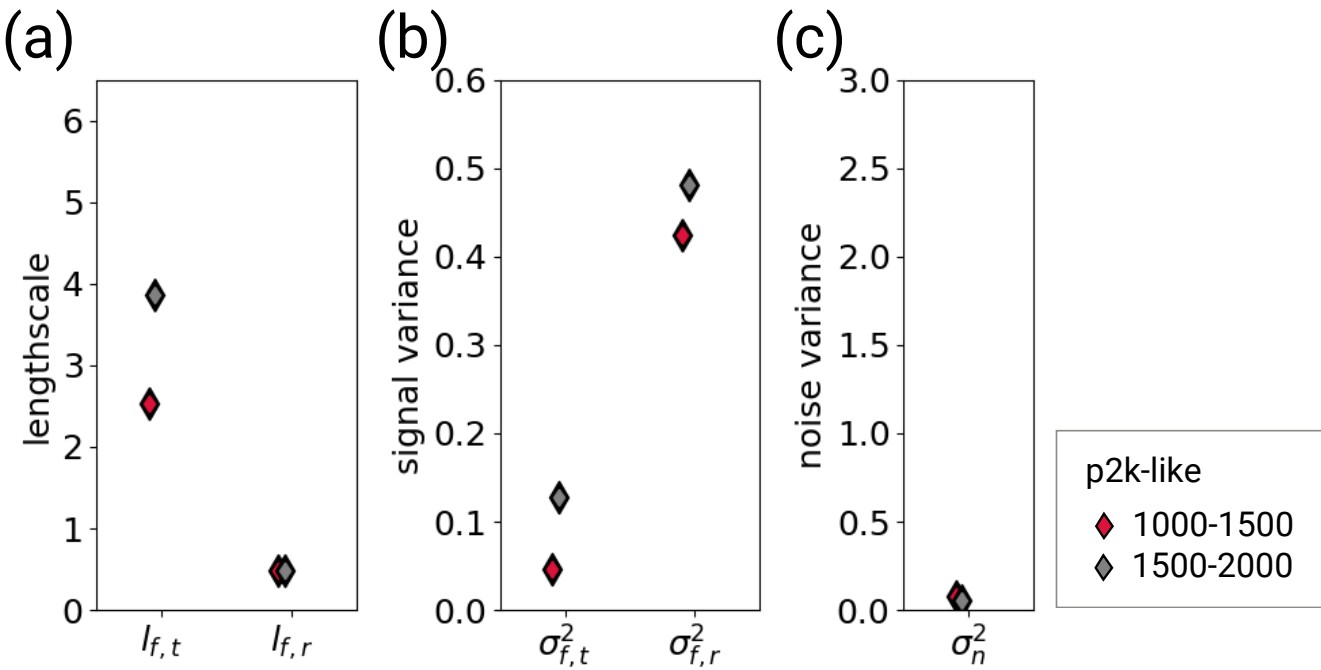

**Figure D3.** Hyperparameters of the sensitivity experiment shown in Fig.D2. Red diamonds correspond to the period 1000-1500, grey diamonds to the period 1500-2000 (grey diamonds here are the same as the red diamonds in Fig. 5). The hyperparameters are **(a)** the typical lengthscales $l_{f,t}$ and $l_{f,r}$, **(b)** the signal variance $\sigma^2_{f,t}$ and $\sigma^2_{f,r}$, and **(c)** the noise variance $\sigma^2_n$. The subscript $t$ indicates that the kernel operates only on the time dimension; the subscript $r$ indicates that the kernel operates on all dimensions, including time (see Eq. 6 and 7). The lengthscales are unitless, corresponding to the unitless distance of the embedding space. The lengthscale $l_{f,t}$ can be transformed into years through division by 1.10. The signal and noise variance are given in $K^2$.

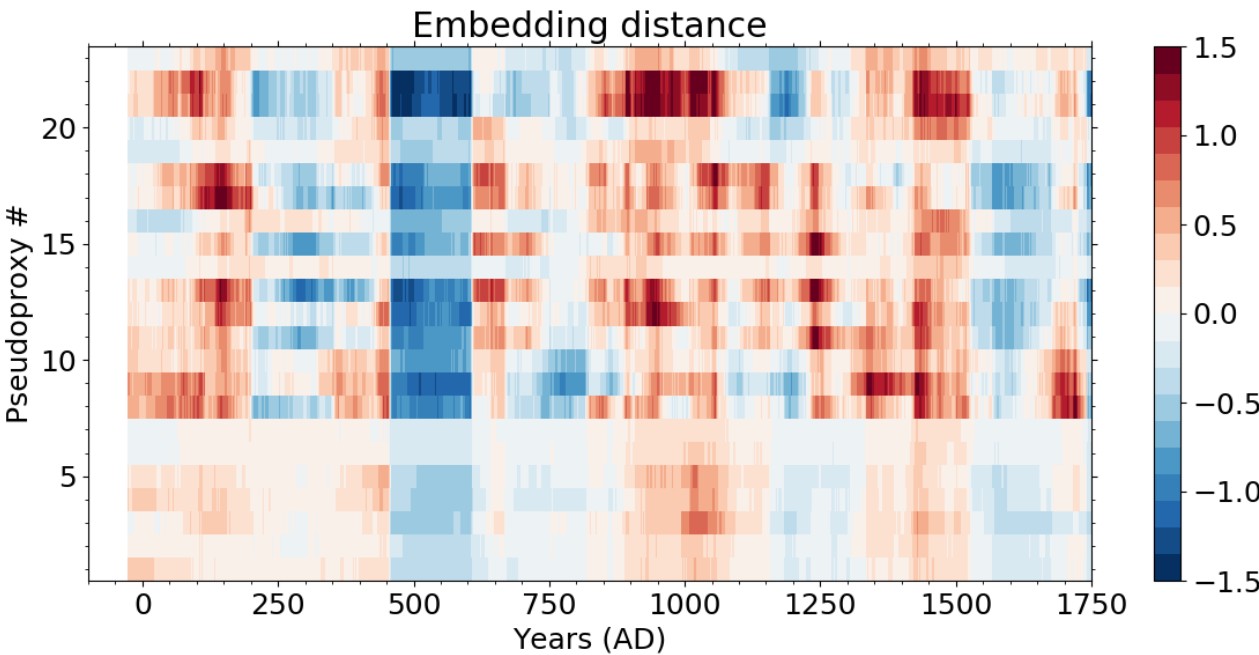

**Figure D4.** Changes of the embedding distance between the AMVI and the pseudoproxy records (Eq. 5) calculated over a running window of 151 years. The anomalies are calculated against the mean distance over the entire period. Blue shading indicates a smaller distance, i.e., more similar records. Red shading indicates a greater difference, i.e., less similar records. The numbers on the y-axis are the correspond to the pseudoproxy numbering in Fig. 1.