# Peer review of "Towards variance-conserving reconstructions of climate indices with Gaussian Process Regression in an embedding space"

_Geoscientific Model Development, 2022_

## Referee Comment (RC1)

**Review of 'Towards variance-conserving reconstructions of climate indices with Gaussian Process Regression in an embedding space'**

The manuscript by Klockmann et al. introduces a new technique for climate index reconstructions from proxy data: Gaussian Process Regression (GPR) in a modified input space named embedding space. The new method is compared with classical principle component regression in pseudo-proxy experiments of varying complexity (PPEs). The PPEs suggest that the new method could be superior to established approaches in reconstructing indices with substantial variability on multi-decadal and longer timescales such as the Atlantic Multi-decadal Variability.

The new method and the results are definitely interesting and worthy of publication, albeit I do not exactly see how the manuscript fits into the list of GMD manuscript types. It does not fit into my understanding of the scope for 'model description papers' because while it describes a new method, the implemented model still seems in a rather experimental stage and the README of the code in the Supplement explicitly states "The scripts are taylored to use the provided test data, i.e. they are not written in a general form that would allow to use them with any kind of suitable dataset, yet." This will make it very hard for readers to use the new method outside of the presented PPEs. If it is designed as a model description paper, at least the model must be given an explicit name and version number following the GMD guidelines and some more effort should be put into making the code usable for others. Since the paper develops a new method and is not related to model improvement it is also not directly a 'development and technical paper'. Therefore, I ask the authors to clarify how the manuscript fits into the GMD manuscript types and adapt it accordingly.

In addition to these general considerations, I have a few major issues I kindly ask the authors to address before publication and additional specific comments listed below.

**Major issues:**

1. Introduction: The paragraphs l. 27-79 describe the AMV-related research fairly extensively and in my opinion much longer than necessary for a model description/development paper. In contrast, the final part (l. 80-89), where the new method is introduced, is a bit short to help me understand the authors thought process in selecting and developing the described methods.

2. Sect. 2.2: The model description varies between long descriptions of general GPR theory / modeling options and fairly short parts on the selected solutions, the motivation behind these choices, and implementation details. I would prefer to focus more on the specific choices for reconstructing the AMVI and why these choices are made. For more specific questions arising from the method description see below.

3. How is $\sigma_n$ handled for the AMVI? Is the AMVI just given by f(z) or is $\varepsilon$ also added in observed and reconstructed AMVI?

4. Sect. 3.2 / 3.3: In the evaluation of the PPEs with noisy pseudo-proxies, the authors focus on the ensemble mean time series across all randomized experiments. I do not see why this is a useful quantity for evaluation. It is not a quantity occurring in reality since the authors correctly state that in real-world applications we only have one realization of pseudo-proxies. Therefore, the ensemble members should be evaluated separately (as each of them is a single realization which could occur in reality) and then the mean and spread of the evaluation measures should be reported and analyzed.

5. As the authors state, GPR is a Bayesian method. Thus, it naturally produces uncertainty estimates through sampling of the posterior distribution. Currently, only the posterior mean is evaluated throughout the manuscript if I see it correctly. Uncertainty quantification is an important part of climate index reconstructions and has been the subject of intense debates over the last decades. Therefore, I would like to see some evaluation on how useful the uncertainty estimates provided by the posterior distribution are.

6. Sect. 3/4: While the authors report several situations where one or several of the reconstruction methods give unreasonable results or fail to reconstruct the underlying truth, explanations for why the models are better in some aspects and worse in others are mostly missing. The authors speculate on some potential reasons but I would like to see some more sensitivity tests to give the reader a better feel for the strengths and weaknesses of the different methods.

**Specific comments:**

- l. 12: Please reformulate one of the 'relevants'

- l. 17: The last 1000-2000 years are normally named the 'Common Era' and not the 'preindustrial period'. 'Preindustrial' could lead to confusion with simulations using fixed preindustrial boundary conditions. Focusing here on the Common Era as a 'must' seems a bit arbitrary since also other periods would be of interest.

- l. 58: Which non-linear methods have shown promise? Is non-linearity really the main advantage here or could other factors also be important?

- l. 103-105: Which climate variable do you use to construct the pseudo-proxies (e.g. SST, surface temperature, near-surface air temperature)?

- l. 109-111: How do the SNR and the construction of pseudo-proxies compare to other pseudo-proxy studies?

- Figure 1a: From the color scale in (a), it is very difficult to distinguish the correlation of the different records. Maybe you can improve the color scale.

- Figure 1c-e: Over which period are the cross-correlation, STD radio, and embedding distance computed?

- Figure 1d: The $10^0$ could be removed.

- l. 132: Are Matern functions really 'very complex' kernels?

- l. 139-140: The inference strategy is described very briefly here. Some more explanation could be useful for readers not that familiar with Bayesian inference

- l. 160: Is there a reason why the abbreviation SVGP is not adopted in the manuscript and 'sparse GP' is used instead?

- l. 163: What is the 'Adam Optimiser'?

- l. 171: How is the AMVI formulated in proxy space since it is defined over a different spatial scale than the individual proxies and how does GPR-based climate index reconstruction work when the GP is formulated as function of the proxy values (=temperature?)?

- l. 177-192: I struggled to understand why the embedding space needs to have the given dimension and the idea of how the embedding space is constructed did not become clear to me until I finished reading Sect. 2.2.3.

- l. 203-205: What are the properties that need to be fulfilled by the distance matrix / distance measure? Why is a positive correlation between records needed between the records?

- l. 209: I guess that equidistant coordinates perform worse because than all records influence the AMVI roughly equally whereas for CC-based coordinates records with a high correlation with the AMVI become more important. But why does including the SR improve the result compared to just using CC?

- Equation (3): Is RA = SR?

- l. 213-218: The description was a bit short here for me to really understand what is happening and why.

- Equations (4/5): The Gaussian kernel functions lead to very smooth (infinitely differentiable) functions, likely much smoother than most processes actually observed in climatology. Does this lead to overly smooth predictions on certain timescales and did you test the procedure with kernels that lead to less smooth posteriors? It also might be useful to write down the final covariance model as an equation.

- l. 230: Is $\sigma_n^2$ the same as a nugget effect in statistical modeling? If so, it might make sense to mention it here for the statistically-inclined readers.

- l. 235/236: Do you have an explanation for why this slightly unusual formulation (two kernels acting in time but only one kernel acts in the "embedding space/distance") performs better than models without $k1$ or with separated kernels acting in time and embedding distance? Does this indicate that the system (AMVI) is better described by two characteristic timescales instead of one similar to two-box energy balance models outperforming one-box energy balance models in predicting sea surface temperatures?

- Equations (6) - (9) did not help my understanding. Either they should be embedded better in the text / explained better or they could be removed.

- l. 259: There is a typo in 'obtain'

- l. 299/300 (and similar parts in subsequent sections): The difference between the MPI-ESM- and CCSM4-based results could be explored a bit further. What are the main differences between the simulations that might explain the differing behavior of the reconstruction methods?

- l. 313/314: Is there an explanation for why the sparse GP performs better for noisy than for perfect pseudo-proxies on multi-decadal timescales?

- Fig. 4: This is an interesting figure to explain some of the differences between methods displayed in Fig. 2/3 but unless I missed something, it is barely discussed in the text.

- l. 359: Could the overestimation of variability in periods with few available proxy records be explained by relying too strongly on a small number of proxy records which tend to be more variable than the AMVI due to integrating over a smaller spatial scale? This could maybe be tested by comparing hyperparameters fitted separately for periods with high and low record availability.

- l. 393-398: Is the introduction of a 'noise variance' parameter similar to error-in-variables approaches for frequentist regression models?

- l. 408: Is the model really using one tenth of the available training data if you use every tenth time step but also an (optimized) subset of the original locations?

- l. 416-418: Can you expand on these length scales and magnitudes? What are expected values and where do the parameters rank in the range of reasonable values?

- l. 444: The GPR-model seems to handle white noise proxies very well, in parts due to the inclusion of the parameter $\sigma_n$. What would happen if the proxy noise would be auto-correlated? Is there a way to adapt the model accordingly?

- Conclusions: Since this paper develops and tests a new methods for climate index reconstructions, it would be very useful for the reader to get some more guidelines for future applications of the method and how it might be applied to other indices.

---

## Author Comment (AC1)

**Reviewer #1**

We would like to thank Referee 1 for their detailed and constructive comments. In the following to explain how we plan to revise the manuscript to address their suggestions.

**The original comments are written in bold font**, our responses with normal font

**General comment**

**C1 The new method and the results are definitely interesting and worthy of publication, albeit I do not exactly see how the manuscript fits into the list of GMD manuscript types. It does not fit into my understanding of the scope for 'model description papers' because while it describes a new method, the implemented model still seems in a rather experimental stage and the README of the code in the Supplement explicitly states "The scripts are taylored to use the provided test data, i.e. they are not written in a general form that would allow to use them with any kind of suitable dataset, yet." This will make it very hard for readers to use the new method outside of the presented PPEs. If it is designed as a model description paper, at least the model must be given an explicit name and version number following the GMD guidelines and some more effort should be put into making the code usable for others. Since the paper develops a new method and is not related to model improvement it is also not directly a 'development and technical paper'. Therefore, I ask the authors to clarify how the manuscript fits into the GMD manuscript types and adapt it accordingly.**

R1 This is to some extent a matter of perspective, and we assume that the editor has already perused the manuscript at the submission stage to assess its suitability to the journal. However, we agree to some extent with the reviewer and we will characterize the method more specifically according to its objectives and methodological steps, as also outlined below in response to the specific comments. Indeed the method cannot be totally universal, but it can certainly find applications in other areas of science and technology, where the objective may be to provide a complete field (e.g., image) from sparse information.

Browsing the journal, we find other manuscript that describe a methodological advance but that are not ripe for a general application. Those manuscripts are, as ours, refinements or combinations of statistical methodologies for a particular purpose. We therefore think that the manuscript does fit into the category of "development and technical paper".

**2. Major issues**

1. **Introduction: The paragraphs l. 27-79 describe the AMV-related research fairly extensively and in my opinion much longer than necessary for a model description/development paper. In contrast, the final part (l. 80-89), where the new method is introduced, is a bit short to help me understand the authors thought process in selecting and developing the described methods.**

We will rebalance the space devoted to the physical motivation and the methodological aspects, placing the model against a broader backdrop of estimation of time series from partial information.

2. C. **Sect. 2.2: The model description varies between long descriptions of general GPR theory / modeling options and fairly short parts on the selected solutions, the motivation behind these choices, and implementation details. I would prefer to focus more on the specific choices for reconstructing the AMVI and why these choices are made. For more specific questions arising from the method description see below.**

R. We will try to strike the right balance between the general GPR description and this particular application. GPR has not been used so far for climate reconstructions and the interested reader might

find it useful to have in this text an introduction about the basic method set-up. Therefore, we think this part, although shortened, can be useful for the reader. We will expand the justification for our particular choices. This justification is mainly based on the need to produce reconstructed time series with the correct amplitude of variability.

**C 3. How is σn handled for the AMVI? Is the AMVI just given by f(z) or is ε also added in observed and reconstructed AMVI?**

R.   The target /reconstructed AMVI is indeed the mean of the GP-posterior, the 'best' estimation of the true value. A realization of sigma_n is not added to the posterior mean. We will include a sentence to clarify this point.   Sigma_n is estimated during training. As we estimate only one sigma_n across all dimensions, it is the same for all proxy timeseries and the target AMVI. This is of course a simplification but the estimated hyperparameters (Fig.4) show that the estimated sigma_n corresponds to the mean noise across all records in most cases and is therefore a good first approximation.

**C 4.  Sect. 3.2 / 3.3: In the evaluation of the PPEs with noisy pseudo-proxies, the authors focus on the ensemble mean time series across all randomized experiments. I do not see why this is a useful quantity for evaluation. It is not a quantity occurring in reality since the authors correctly state that in real-world applications we only have one realization of pseudo-proxies. Therefore, the ensemble members should be evaluated separately (as each of them is a single realization which could occur in reality) and then the mean and spread of the evaluation measures should be reported and analyzed.**

R. As per a comment by reviewer #2 We will present summaries of the probability distributions of the evaluation metrics for the different model set ups.   This information is already available  for the MPI-ESM based Tcnpp ensemble in the appendix (Fig.B1). Note that also the min-max range of the ensemble spread is already given for every ensemble metric throughout the text. We will modify the text to focus on the mean and spread of the metrics. This will however not change our conclusions about the respective performances (see Fig.B1).

**C  5. As the authors state, GPR is a Bayesian method. Thus, it naturally produces uncertainty estimates through sampling of the posterior distribution. Currently, only the posterior mean is evaluated throughout the manuscript if I see it correctly. Uncertainty quantification is an important part of climate index reconstructions and has been the subject of intense debates over the last decades. Therefore, I would like to see some evaluation on how useful the uncertainty estimates provided by the posterior distribution are.**

R The manuscript in its original form does address, albeit partially, the uncertainty in the estimation of important hyperparameters, such as the local noise represented by sigma_n, and the scale  parameters in the kernel. This is represented in Figure 4. We acknowledge that the figure caption is not clear enough in this respect and we will state this point more clearly.  The confidence intervals given in Fig. 2 and 3 correspond to the posterior uncertainty (2sigma of the posterior distribution). The displayed uncertainty ranges in Fig.3 are an average of the 2sigma ranges of  the individual ensemble members.
We will also evaluate whether the true AMV index does lie within the 95%- uncertainty ranges in 95% of the time steps, or whether the GPR-derived uncertainty ranges too liberal or too conservative are. We thank the reviewer for this good point.

**C 6. Sect. 3/4: While the authors report several situations where one or several of the reconstruction methods give unreasonable results or fail to reconstruct the underlying truth, explanations for why the models are better in some aspects and worse in others are mostly missing. The authors speculate**

**on some potential reasons but I would like to see some more sensitivity tests to give the reader a better feel for the strengths and weaknesses of the different methods.**

R. We have tried in the original manuscript to interpret the results obtained with the different set-ups of the GPR model, but at some point its difficult to point out to particular reasons for their behaviour, as it happens in many other applications of machine learning. Perhaps the most relevant interpretation is why the different behaviour in the skill of the Sparse GPR and standard GPR with respect to the presence of noise in the proxies. One assumption, as suggested by the two reviewers, is that the number of proxy records and the presence of noise in the proxy records may affect the overfitting tendency of the model. we will test this hypothesis by conducting a targeted experiment with larger and smaller proxy networks.

**Specific comments**
**l. 12: Please reformulate one of the 'relevants'**
We believe the use of "relevant" to be fine in both cases.

● **l. 17: The last 1000-2000 years are normally named the 'Common Era' and not the 'preindustrial period'. 'Preindustrial' could lead to confusion with simulations using fixed preindustrial boundary conditions. Focusing here on the Common Era as a 'must' seems a bit arbitrary since also other periods would be of interest.**
R. We will change pre-industrial period to Common Era

● **l. 58: Which non-linear methods have shown promise? Is non-linearity really the main advantage here or could other factors also be important?**
R. Data assimilation methods based on k-nearest neighbor (Pfister et al., 2020 doi:10.5194/cp-16-663-2020). However, these require the use of climate simulations. Also random forest (Michel et al 2020, https://doi.org/10.5194/gmd-13-841-2020). We will include a brief disscussion of these methods in the introduction.

● **l. 103-105: Which climate variable do you use to construct the pseudo-proxies (e.g. SST, surface temperature, near-surface air temperature)?**
R. Surface-air temperature. This is the standard variable that for instance temperature sensitive tree-rings represent. As noted in line 100.

● **l. 109-111: How do the SNR and the construction of pseudo-proxies compare to other pseudo-proxy studies?**
The amount of local interannual noise in real proxies usually assessed by the correlation between the proxy time series and the instrumental time series. These correlations may be in the range 03 to 0.7. The amount of local noise used in other pseudo-proxy studies is within this range, as ours. We will include a sentence in the revised version.

● **Figure 1a: From the color scale in (a), it is very difficult to distinguish the correlation of the different records. Maybe you can improve the color scale.**
We chose this color scale to match the one used for the cross-correlation matrix in Fig.1c. But we agree, the respective values are not easily distinguished in this case. We will use a non-continuous color scale, so that the correlations can be more easily inferred by eye.

● **Figure 1c-e: Over which period are the cross-correlation, STD radio, and embedding distance computed?**
For the pseudoproxy records the entire 2000 years of the simulation, for all pairs including the AMVI only the most recent 150 years. We will add this information in Section 2.2.3.

● **Figure 1d: The 100 could be removed.**
Noted and it will be amended

● **l. 132: Are Matern functions really 'very complex' kernels?**
We will change to 'more complex kernels'

● **l. 139-140: The inference strategy is described very briefly here. Some more explanation could be useful for readers not that familiar with Bayesian inference**
We will add more explanation about inference

● **l. 160: Is there a reason why the abbreviation SVGP is not adopted in the manuscript and 'sparse GP' is used instead?**
This was done because we later only use "full and sparse *em*GP" when we refer to the GP in embedding space. In section 2.2.1 we will stick instead to SVGP. In the remainder of the manuscript we will stick to the full and sparse emGP nomenclature.

● **l. 163: What is the 'Adam Optimiser'?**
R. The Adam optimiser is an algorithm to search the best model parameters according to a prescribed cost function. It belongs to the family of stochastic gradient descent algorithms and it is widely used in machine learning applications due to is favorable properties. Essentially, it takes into account the mean and standard deviations of the gradient of the cost function in previous optimization iterations to propose a new value of the parameters. We will include this sentence to complement the reference.

● **l. 171: How is the AMVI formulated in proxy space since it is defined over a different spatial scale than the individual proxies and how does GPR-based climate index reconstruction work when the GP is formulated as function of the proxy values (=temperature?)?**
R. We interpret 'spatial scale' in this comment as 'amplitude of variability' In this particular application, the pseudo-proxies and the AMV index are all defined as near-surface air temperature, so that the range of variability is roughly similar for all of them. The reviewer is right that this condition cannot be generalized and for other reconstructions the proxies and target time series would need to be standarized to unit variance. We will include a brief explanation in this regard.
.

● **l. 177-192: I struggled to understand why the embedding space needs to have the given dimension and the idea of how the embedding space is constructed did not become clear to me until I finished reading Sect. 2.2.3.**
R. Indeed the new set-up of the GPR is not easy to visualize. We will reformulate this important part of the manuscript more clearly. The number of necessary dimensions is most easily conceived from the easiest setup with equal distances between all possible pairs of records. Imagine a hypothetical case with four records (e.g. three proxy records and one AMVI). To be able to arrange all four records such that there is an equal distance between all respective pairs of records, one needs a three dimensional

embedding space. Thus for q timeseries, the embedding space must have a dimension of q-1. Time then is added as an additional dimension.

● l. 203-205: What are the properties that need to be fulfilled by the distance matrix / distance measure? Why is a positive correlation between records needed between the records?
R. In our set-up, the distance matrix does not require any specific properties, other than that the distance should be a monotonous function of the 'similarity' between time series and symmetric. In our case we have chosen the correlation between the time series as a measure of distance, augmented by the amplitude of variations, both combined so that the resulting metric is symmetric. We think this is a reasonable choice.

● l. 209: I guess that equidistant coordinates perform worse because than all records influence the AMVI roughly equally whereas for CC-based coordinates records with a high correlation with the AMVI become more important. But why does including the SR improve the result compared to just using CC?
R. The reviewer is correct about the case with the equidistant coordinates, When performing the reconstructions only with CC-based coordinates, we found that the reconstruction is dominated by northern hemisphere records which have a much larger range of variability. This became especially important for networks with realistic proxy availability. Further back in time, only NH records are available and the reconstructed variability for earlier periods was strongly overestimated in the CC-based case. The consideration of the SR in the metric ensures that records that may have larger variability are considered farther away from the target, this improved the magnitude if the reconstructed variability. An alternative approach would be the normalization of all records. In that case, CC-based coordinates would be sufficient.

● Equation (3): Is RA = SR?
R Yes, it will be corrected

● l. 213-218: The description was a bit short here for me to really understand what is happening and why.
R. The normalisation of the time axis is done to ensure that the variations along the time axis are comparable to the variations along the other embedding dimensions. This is a necessary step due to how the kernel k2 is formulated, distance in time and embeddin pace must comparable, to allow for interactions across tim and records. Without the normalisation, either the distance between records or the timescale would dominate the lengthscale of k2. If the kernel would be formulated such that time and embedding dimensions were treated separately, this normalisation would likely not be necessary. But as stated in the manuscript, the current kernel formulation outperforms the separated kernel. We will explain this point more clearly.

● Equations (4/5): The Gaussian kernel functions lead to very smooth (infinitely differentiable) functions, likely much smoother than most processes actually observed in climatology. Does this lead to overly smooth predictions on certain timescales and did you test the procedure with kernels that lead to less smooth posteriors? It also might be useful to write down the final covariance model as an equation.
R. The temporal component of Gaussian kernel leads to predictions that are indeed smooth in time, as it acts as a low-pass filter on the predictor time series. However, this does not mean that the model is

not able to represent rapid changes of the target variable if the proxy records also change rapidly. The case that the reviewer raising - a non differentiable temporal behavior- is in our opinion extremely rare. We will also complement equations 4 and 5 with a full expression of the covariance model.

● l. 230: Is σn 2 the same as a nugget effect in statistical modeling? If so, it might make sense to mention it here for the statistically-inclined readers.
R. Yes, we thank the review for this suggestion.

● l. 235/236: Do you have an explanation for why this slightly unusual formulation (two kernels acting in time but only one kernel acts in the "embedding space/distance") performs better than models without k1 or with separated kernels acting in time and embedding distance? Does this indicate that the system (AMVI) is better described by two characteristic timescales instead of one similar to two-box energy balance models outperforming one-box energy balance models in predicting sea surface temperatures?
R The rationale is to allow for different time scales in the autocorrelation (k1) and in the cross-correlation between proxy records (k2). Also, the lengthscale of k2 cannot be interpreted as a pure timescale as it discribes the typical lengthscale for the influence of the respective records across time and embedding space. We will explain this point more clearly. We think that this is not related with the issue raised by the reviewer, as our model does not include any type of forcing or reservoirs for the Atlantic Ocean. If we had included in the predictors a forcing time series, then the interpretation of the time scale would indeed be closely related to a thermal inertial timescale of the ocean.

● Equations (6) - (9) did not help my understanding. Either they should be embedded better in the text / explained better or they could be removed.
R These equations were meant to better explain the embedding process, but perhaps they should be better introduced. We will also formulate these equations in symbolic terms as product of matrix and vectors.

● l. 259: There is a typo in 'obtain'
R Noted

● l. 299/300 (and similar parts in subsequent sections): The difference between the MPI- ESM- and CCSM4-based results could be explored a bit further. What are the main differences between the simulations that might explain the differing behavior of the reconstruction methods?
R. We suspect that the main reason must be the different spatial correlations within the temperature field. We will explore this in terms of the spatial variability modes (e.g. EOFs)

● l. 313/314: Is there an explanation for why the sparse GP performs better for noisy than for perfect pseudo-proxies on multi-decadal timescales?
R. This is an interesting question (also posed by reviewer #2) , but it is not easy to address. We assume that the sparse GP is less impacted by local noise when using only a portion of the data available for the training, and thus it can behave differently considering overfitting. As mentioned before, we will include a targeted pair of experiments with a large and asmall proxy network.

● **Fig. 4: This is an interesting figure to explain some of the differences between methods displayed in Fig. 2/3 but unless I missed something, it is barely discussed in the text.**

R. Actually large parts of the discussion refers to Fig 4 but we will make sure to discuss Fig.4 even more deeply.

● **l. 359: Could the overestimation of variability in periods with few available proxy records be explained by relying too strongly on a small number of proxy records which tend to be more variable than the AMVI due to integrating over a smaller spatial scale? This could maybe be tested by comparing hyperparameters fitted separately for periods with high and low record availability.**

R. This is certainly a reasonable explanation. In fact, dendroclimatologists apply a statistical tool specially designed for this purpose called 'variance stabilization'. Our objective is, however, to compare the methods in different situations , i.e. is method A or B closer to the truth, and not so much the correction of the effect of sparser proxy networks. That would be a different methodological paper.

In our specific case, however, we think that the suggestion of the reviewer can only be part of the explanation. On the one hand we try to amend for the effect of few records wtwithh large variability through the additional SR scaling in the distance matrix. On the other hand, if this were the only explanation, we would expect the effect to get stronger the further we go back in time as the number of records decreases further. An additional explanation could also be non-stationary cross-correlations between records, this is however difficult to test for. Nonetheless, as we are planning to conduct experiments with also a smaller number of records in response to other comments regarding overfitting, we will also compare hyperparameters for the different data availability, as suggested by the reviewer.

● **l. 393-398: Is the introduction of a 'noise variance' parameter similar to error-in-variables approaches for frequentist regression models?**

R. Yes, it is conceptually similar in the sense that uncertainty in the 'predictors' is also taken into consideration. The EIV model, however, requires the knowledge of the ratio of noise in the independent and dependent variables, whereas here the noise variance is estimated along with with hyperparameters, We will refer to the EIV model to make this link explicit to the reader.

● **l. 408: Is the model really using one tenth of the available training data if you use every tenth time step but also an (optimized) subset of the original locations?**

During the entire optimisation/fitting procedure, the model always uses a slightly different subset in every optimisation step, so in a sense it uses more than a tenth of the data<s>probably uses more than that</s>. But the resulting co-variance matrix is based on an optimised subset which always corresponds to a tenth of the data.

● **l. 416-418: Can you expand on these length scales and magnitudes? What are expected values and where do the parameters rank in the range of reasonable values?**

R. We will expand the discussion on the value of the hyperparameters, as per comment on Figure 4

● **l. 444: The GPR-model seems to handle white noise proxies very well, in parts due to the inclusion of the parameter $\sigma_n$. What would happen if the proxy noise would be auto-correlated? Is there a way to adapt the model accordingly?**

R. We are not aware of an application of of GPR for the case the reviewer is suggesting, but the GPR model could be modified to account for temporal autocorrelation in sigma_n. Actually, it seems that the

GPR model would need to be cast in an embedding scape similar to the one implemented here (explicitly including the temporal dimension). This augmented model could be even expanded to account for spatial and temporal correlation of the sigma_n - in that case sigma_n would not be just a a random variable but a gaussian random field itself, described by three additional hyperparameters (the local noise, one temporal decorrelation and one spatial decorrelation). The calibration would become certainly more complex.

The case of spatial correlation of the proxy noise would be somewhat special, and not that common for paleoclimate reconstructions, but reasonable for certain type of proxies, so we will briefly also discuss this possibility in the revised version.

● **Conclusions: Since this paper develops and tests a new methods for climate index reconstructions, it would be very useful for the reader to get some more guidelines for future applications of the method and how it might be applied to other indices.**

**R.** The current GPR model is applicable with slight modifications in a broad set of situations in paleoclimate, in which just one index is reconstructed. The reviewer has hinted in other comments at other possible situations in which the GPR method does require a more complex model. For instance, in the case of reconstructions of a spatially resolved field or with autocorrrelated noise terms. There are still others that we have not mentioned, like the use of proxy records of different nature - e.g. tree-rings and lake sediments, which display different statistical properties.

We will expand the discussion and the conclusions to present those cases.

**Reviewer #2**

We would like to thank Referee 2 for their detailed and constructive comments. In the following to explain how we plan to revise the manuscript to address their suggestions.

**The original comments are written in bold font**, our responses with normal font

**C The quality of the paper is generally high, the analysis appears accurate from my knowledge, and there are no major points which I think should prevent its publication. There are however several minor points and suggestions which I identified and reported below, which I think could improve the paper further.**
R. We thank the reviewer for the general positive assessment

**Lines 33-34: "however" is awkward in the middle of the negation.**
Noted

**C Line 134: Might be useful to describe the term 'hyperparameter' for those outside the machine learning field, how is it different from a normal 'parameter'?**
R. The reviewer is right. This notation is usual in the Gaussian Process literature, but confusing outside the machine learning community

**C.Lines 148: So the batch size is the total number of observations across records (i.e. 5 records with 100 observations plus 2 records with 200 observations would mean a batch size of 900)?**
R. The batch size corresponds to number of training observations given to the algorithm in. In our case it is defined as suggested by the reviewer.

**C. Lines 180-183: How are the irregular resolution of the proxies handled (for the realistic P2k case)? Or are the pseudo-proxies created at annual resolution? In which case it would be helpful to briefly hint in the discussion how realistic irregular proxies could be used in the future and how it might affect the results.**
R. The case considered here is when all proxies are annually resolved, and all have similar statistical properties (differing only in their amplitude of variability). the case of proxies with different statistical properties is much harder to handle with a GPR model. Assuming that the resolution is known, it would need to be incorporated in the Gaussian process prior, and the inference of the posterior will not be given by the standard equations any more. In this case, it seems to us that the more immediate approach would be to subsample all records according to the one with least resolution, although much information would be lost. This is the standard approach for long time scale paleoreconstructions. Other possibility is to set up two GPR models, one for the records with high-frequency resolution, applying a high-pass filter to those) and one GPR model for the records with low-frequency resolution (low-pass filtering the records with high resolution). This approach is sometimes also used in paleoclimatology, but one cveat is the difficulty to calibrate a model for low-resolution records, since the observational period is too short. We believe that this difficulty is common to all methods, and not a unique faeture of GPR.
We will include this point in the discussion.

**C. Equation 3: Should RA be SR?**
Yes, it will be corrected.

**C. Lines 248-249: So the embedding coordinates are calculated from the distance matrix via multidimensional scaling? Might not be obvious for the layman (including me) how the matrix is obtained, could it be made more explicit in an Appendix?**

Yes, the reviewer is correct. We will explain this point more clearly.

**C Line 259: "obatain"**

R Noted

**C Equation 6,7,8,9: Shouldn't the matrix be equal to something for an equation? Make explicit which one is the input and which the output.**

R . Yes, we will include a version of this equation in a more symbolic form

**C Lines 299-300: Could it be because CCSM4 is more homogeneous (less spatial degrees of freedom) since the mean embedding distance between the records is smaller? Just a thought.**

R. This is certainly a possibility. As per a comment by the reviewer one, we will explore the degrees of freedom in both models , for instance by an EOF analysis., and look at the structure and length scales of the resulting spatial patterns.

**C Figure 2: Unclear to me what the 95% confidence intervals represent in the temporal domain? Are they the spread of the unsmoothed data? Also for the spectra it is not explained what the confidence intervals are, simply chi-square CI with 2 degrees of freedom? It is a question of style and not necessary, but I personally like smoothing the spectra to make for a clearer comparison (e.g. Using a Gaussian smoothing kernel as in JW Kirchner, Aliasing in 1/f(α) noise spectra: Origins, consequences, and remedies. Phys Rev E Stat Nonlin Soft Matter Phys 71, 066110; 2005).**

R We will explain this point more clearly. The uncertainty ranges correspond to the 2 sigma range of the posterior GP distribution.. The ranges displayed in the Figure are an average over all the ensemble members. The Spectral uncertainty given for the target spectrum and the spectrum of the mean does indeed correspond to the chi-square CI.

**C. Caption Figure 2 and 3: "powerspectra" -> "power spectra"**

R Noted

**C Figure 3: I don't understand the indicated 95% CI number. Do those correspond to the same CI shown on Figure 2?**

R Yes, we will make it explicit

**C Section 3.2: I would generally favour calculating the statistics for individual ensemble members and reporting the mean+/- standard deviation rather than calculating them with respect to the ensemble mean. Similarly, I would show the mean of the spectra rather than the spectrum of the mean for Figure 3 b,d,f as it is more representative of the real result one would obtain.**

R. This information is partly already available for the MPI-ESM based Tcnpp ensemble in the appendix (Fig.B1). Note that also the min-max range of the ensemble spread is already given for every ensemble metric throughout the text. We will modify the text to focus on the mean and spread of the metrics. This will however not change our conclusions about the respective performances (see Fig.B1). We will also modify the spectrum figures to show the mean of the spectra rather than the spectrum of the mean.

**C Figure 3 b,d,f: I wonder whether the PCR has the right high-frequency amplitude for the**

**right reason? Are the high-frequencies just noise and thus PCR doesn't perform better than the other methods or are they actually correlated with the real series?**

R This a behaviour that has been found in other previous analysis. Yes, the PCR reconstructions are indeed correlated with the target at high frequencies.

**C Lines 329-331: Do the authors have an idea why the sparse can outperform the full GP? Could it be a case of overfitting when noise is present? Such that the sparse one is less sensitive to overfitting?**

R This is an interesting question (also posed by reviewer #1). The suggestion by the reviewer is a reasonable explanation, As mentioned in previous comments, we will perform a pair of targeted experiments with a large and a small proxy network to test this hypothesis.

**C Lines 335-339: How are data resolution handled? If there is 5 years resolution, then are there gaps between the years or are the values interpolated? Or are the annual data used and only clipped at the end of the record?**

R The latter case. In the manuscript only proxies with annual resolution are created and clipped at the end of the record.. This is the standard case for past past millennium reconstructions. The case with proxies with different temporal resolution is much more difficult to handle (see previous comment). We will include a discussion on this case, but we will not be able to find an explicit solution.

**C Line 387: I would remind the reader for the discussion what AMV-relevant timescales are. Maybe write in parenthesis something like (decadal to multi-decadal).**

R. Noted

**C One issue I would like better discussed is the loss of variance on longer than centennial timescales in the sparse emGPR for the full 2k run. To me this is quite an important limitation since I don't think it makes sense to restrict AMV-relevant timescales to decadal to multi-decadal; there is a continuum of processes and I don't think there are reasons to believe that it would flatten out on longer than centennial timescales or be related to a separate non-AMV relevant process right?**

R. The length of the available simulations is 1000-2000 years, so that it is difficult to assess the behavior on multicentennial timescales - the degrees of freedom is considerably reduced. For instance, the correlation between reconstruction and target woud be estimated using, say, 10 degrees of freedom, assuming that centuries are independent samples.

**C Line 465-466: Looking forward to seeing how it compares to the traditional PCR method!**

R. We are also curious to see how it will compare to existing reconstructions. This is to be the focus of a follow up study.

---

## Author Response (AR1)

**Reviewer #1**

We would like to thank Referee 1 for their detailed and constructive comments. In the following we explain how we revised the manuscript to address their suggestions.

**The original comments are written in bold font**, our responses with normal font and in colour. Modified text passages are highlighted with *italics.*

**1. General comment**

**C1 The new method and the results are definitely interesting and worthy of publication, albeit I do not exactly see how the manuscript fits into the list of GMD manuscript types. It does not fit into my understanding of the scope for 'model description papers' because while it describes a new method, the implemented model still seems in a rather experimental stage and the README of the code in the Supplement explicitly states "The scripts are taylored to use the provided test data, i.e. they are not written in a general form that would allow to use them with any kind of suitable dataset, yet." This will make it very hard for readers to use the new method outside of the presented PPEs. If it is designed as a model description paper, at least the model must be given an explicit name and version number following the GMD guidelines and some more effort should be put into making the code usable for others. Since the paper develops a new method and is not related to model improvement it is also not directly a 'development and technical paper'. Therefore, I ask the authors to clarify how the manuscript fits into the GMD manuscript types and adapt it accordingly.**

R1 This is to some extent a matter of perspective, and we assume that the editor has already perused the manuscript at the submission stage to assess its suitability to the journal. However, we have characterised the method more specifically according to its objectives and methodological steps, as also outlined below in response to the specific comments. Indeed the method cannot be totally universal, but it can certainly find applications in other areas of science and technology, where the objective may be to provide a complete field (e.g., image) from sparse information.

Browsing the journal, we find other manuscripts that describe a methodological advance but that are not ripe for a general application. Those manuscripts are, as ours, refinements or combinations of statistical methodologies for a particular purpose. We therefore think that the manuscript does fit into the category of "development and technical paper".

**2. Major issues**

**C 1 Introduction: The paragraphs l. 27-79 describe the AMV-related research fairly extensively and in my opinion much longer than necessary for a model description/development paper. In contrast, the final part (l. 80-89), where the new method is introduced, is a bit short to help me understand the authors thought process in selecting and developing the described methods.**

R. We have shortened the AMV-related part and devoted more space to motivate our choice of method. We refer the reviewer to the new version of the manuscript for the reformulated introduction.

**C 2. Sect. 2.2: The model description varies between long descriptions of general GPR theory / modeling options and fairly short parts on the selected solutions, the motivation behind these choices, and implementation details. I would prefer to focus more on the specific choices for reconstructing the AMVI and why these choices are made. For more specific questions arising from the method description see below.**

We would argue that most of the original method section was not about general GPR theory (except section 2.2.1). But in response to this comment, we have restructured and re-written large parts of the method section in order to make our choices and motivations clearer. We have put special emphasis on the motivation for and design of the embedding space and hope that the concept is clearer now. In rewriting, we have also addressed the more specific issues raised by the reviewer below. Please see the manuscript with tracked changes for the modifications.

**C 3. How is $\sigma_n$ handled for the AMVI? Is the AMVI just given by $f(z)$ or is $\varepsilon$ also added in observed and reconstructed AMVI?**

R. The reconstructed AMVI is indeed the mean of the GP-posterior, the 'best' estimation of the true value. A realisation of sigma_n is not added to the observed AMVI nor the posterior mean. Sigma_n is estimated during training. As we estimate only one sigma_n across all dimensions, it is the same for all proxy timeseries and the target AMVI. This is of course a simplification but the estimated hyperparameters (Fig.4) show that the estimated sigma_n corresponds to the mean noise across all records in most cases and is therefore a good first approximation. We have included a short statement in section 2.3.4:

*"The noise variance sigma_n2 is assumed to be the same across all dimensions, i.e., the learned estimate will be the same for all pseudoproxies and the AMVI. This is a simplification, because every pseudoproxy contains its own level of noise. We will show that this simplification is a good first approximation and enables the GPR to handle uncertain pseudoproxies well."*

**C 4. Sect. 3.2 / 3.3: In the evaluation of the PPEs with noisy pseudo-proxies, the authors focus on the ensemble mean time series across all randomized experiments. I do not see why this is a useful quantity for evaluation. It is not a quantity occurring in reality since the authors correctly state that in real-world applications we only have one realization of pseudo-proxies. Therefore, the ensemble members should be evaluated separately (as each of them is a single realization which could occur in reality) and then the mean and spread of the evaluation measures should be reported and analyzed.**

R. Point taken. We have removed the ensemble mean reconstructions, as well as the spectrum of the mean from the figures. Instead of the metrics of the ensemble mean, we present the mean and spread of the metrics from the individual ensemble members. This information was already partly available in the first versions of the manuscript (old Fig. B1 for the MPIESM TCnpp case), but we have now calculated it for all noisy reconstructions and modified all figures and text accordingly. Some of the numbers have slightly changed, but the conclusions have remained the same.

Appendix C now contains the distribution of the skill metrics for all noisy reconstruction ensembles (MPIESM TCnpp, MPIESM TCp2k and CSESM TCnpp).

Text was adjusted in Section 3.2 and the relevant paragraphs of 3.3.1 and 3.3.2.

**C 5. As the authors state, GPR is a Bayesian method. Thus, it naturally produces uncertainty estimates through sampling of the posterior distribution. Currently, only the posterior mean is evaluated throughout the manuscript if I see it correctly. Uncertainty quantification is an important part of climate index reconstructions and has been the subject of intense debates over the last decades. Therefore, I would like to see some evaluation on how useful the uncertainty estimates provided by the posterior distribution are.**

R. It is true that we mostly discuss the posterior mean. However, the posterior uncertainty estimate is included in the originals Figures 2 and 3 in form of the 95% confidence interval (shading in Figure 2 and number in Figure3). The posterior uncertainty estimate is likely too conservative, as the target index lies within the 95% interval 100% of the time. By definition, it would lie within the 95% interval only 95% of the time. We have included short passages on this in Section 3.1:

*"The GP related uncertainty, as given by the 95th percentile of the posterior distribution, is small for the years 1850 to 2000 where the AMVI has been constrained during training. The uncertainty increases for the reconstruction period. Overall, the posterior uncertainty estimate appears a bit too large - i.e., too conservative - because the true AMVI lies always within the 95\% confidence interval."*

And in Section 3.2:

*"The use of noisy pseudoproxies has approximately doubled the width of the 95% confidence intervals for all three methods. The mean uncertainty range over all emGP ensemble members is +/-0.57, which is again too conservative but reasonable given the amount of non-climatic noise in the pseudoproxies. The mean PCR uncertainty range is +/-0.21, which is likely too confident in combination with the large reconstruction bias."*

We have also formulated the captions more clearly to indicate where we show the posterior uncertainty estimates in the figures.

**C 6. Sect. 3/4: While the authors report several situations where one or several of the reconstruction methods give unreasonable results or fail to reconstruct the underlying truth, explanations for why the models are better in some aspects and worse in others are mostly missing. The authors speculate on some potential reasons but I would like to see some more sensitivity tests to give the reader a better feel for the strengths and weaknesses of the different methods.**

R. As the reviewer says, we have tried to come up with potential causes for the overall performance of the respective methods and test cases. At some point, however, it becomes difficult to identify the exact reason for their behaviour, as it happens with many other applications of machine-learning methods. We do not think that we will be able to explain, e.g., why the full emGP in TCppp is able to capture the target AMVI well in all periods except in the period from 1630 to 1680. But we have tried to expand our explanation for the behaviour of the different models and set-ups:

1. Why do the methods perform differently with MPIESM- and CCSM-based pseudoproxies?

   We have performed an EOF analysis of the simulated surface temperature fields from which we generated the pseudoproxies and the AMVI in the respective simulations. We find that CCSM is more spatially coherent in that the first EOF explains a much larger fraction of the variability than in MPIESM (see Figure R1.1). This can explain why the PCR performs much better in the TCppp case with CCSM data. As noted by reviewer 2 (comment to line 299-300), the greater spatial coherence is also reflected in the overall smaller embedding distance. The mean of the distance matrix is 1.44 (maximum is 3.91)  for MPIESM and 1.10 (maximum 3.08) for CSSM4. The fact that the full and sparse emGPR perform about equally well for

MPIESM and CCSM4 indicates that the emGPR is more robust to different degrees of spatial coherence in the underlying field. This can be considered an additional strength of the emGPR.

2. How do the hyperparameters change with decreasing proxy availability (also in answer to comment on 359)?

To address this, we have retrained and fitted the full emGPR with realistic proxy availability for the period 1000-1500 (for the AMVI, we only used years 1350-1500 for training). The skill is improved with the new set of hyperparameters, the correlation increases, the variance is underestimated by 11% instead of overestimated by 25%, and the relative bias decreases (Figure R1.2). The timescale $l\_f,t$ decreases, as well as the signal variance $sigma2\_f,t$ and $sigma2\_f,r$ (see red and grey diamonds Figure R1.3). The lengthscale $l\_f,r$ and the noise variance remain unchanged. This change in hyper parameters makes sense as it reflects the fact that the AMVI variability (and likely also that of the pseudoproxies) is lower in the years 1000-1500 than in the years 1500-2000. So the somewhat overestimated variability in the years 900-1500 in TCp2k with perfect pseudoproxies and realistic proxy availability (fig.5a) could indeed be explained by a set of not exactly fitting hyperparameters. Non-stationarity in the hyperparameters (or model-parameters in general) is something all reconstruction methods have in common and which is difficult to fix with real-world data availability. We try to overcome the problem somewhat in using as much proxy information as possible for training and creating the distance matrix.

3. Why does the sparse emGP work better with noisy pseudoproxies than with perfect pseudoproxies?

One hypothesis here is that the SVGP is more robust against overfitting in the presence of noise. To test this, we repeat the TCppp and TCnpp cases with the sparse emGP but with a proxy network that contains only a subset of eleven randomly selected pseudoproxies from the original 23 MPIESM-basd records (and with a subset of twelve randomly selected pseudoproxies from the original 24 CCSM4-based records). The idea was that if overfitting was the reason for the difference in skill, the sparse emGP will no longer perform better with noisy data if the proxy network is smaller. The reconstructed AMVI can be seen in Figure R1.4. Given that the proxy network size is reduced by half, the overall reconstruction skill is remarkably similar to that with the full networks with both MPIESM and CCSM data. The difference between the reconstruction with perfect and noisy data is reduced with respect to the full networks. The TCppp reconstruction has a comparable reconstruction skill to single TCnpp reconstruction members. We interpret this such, that the better performance of the sparse emGP with noisy data can at least partly be attributed to a greater robustness against overfitting in the presence of noise.

See next point for a discussion on why the sparse emGP might outperform the full emGP in the CCSM4-based TCnpp case.

4. What about non-optimal hyper-parameters?

As already discussed in the original manuscript, we think that the lower reconstruction skill in the CCSM4-based TCnpp case with the full emGP is due to a non-optimal set of hyperparameters. The signal variance $sigma2\_f,r$ and the noise variance $sigma2\_n$ appear switched, so that in this case, the emGP did not correctly recognise the noise and instead assigned the noise variance to signal variance. Simply repeating the reconstruction with the two variances switched improves the skill slightly (not shown). Thus, also the lengthscales might be non-optimal, e,g, a too low $l\_f,r$. Repeating the full emGP reconstruction with the

set of hyperparameters from the sparse emGP gives almost identical results to the sparse emGP TCnpp reconstruction (which could be expected).

5. Are the embedding distances constant in time?

A last possibility for mismatches between the target AMVI and the reconstructed AMVI could be non-stationary embedding distances. Temporary shifts in the similarity could explain why some periods are captured better than others. In Figure R1.6, we show the changes in the embedding metric $D_{ij}$ for the AMVI with all respective MPIESM-based pseudorecords, calculated over a running window of 151 years. This shows how the similarity/distance between the AMVI and the pseudoproxies varies on time. For all AMVI/pseudoproxy pairs the distance varies in time, and there are also bands, where the distance varies almost uniformly across all pairs, e.g. years 450-600 (distance smaller than average) and years 900-1100 (distance larger than average). This means that the impact of some records will sometimes be over- or underestimated by the emGP reconstruction, which can temporarily reduce the reconstruction skill. As with non-stationary hyperparameters, this is something we cannot change, and which is common to all reconstruction methods. By calculating the embedding distance over the entire length of the pseudoproxy records, we try capture the mean distance which best represents the entire reconstruction period.

We have expanded our discussion (Section 4) to include the above insights, and also point to the explanations already in the results section (Section 3). (see revised manuscript with tracked changes)

**3. Specific comments**

**l. 12: Please reformulate one of the 'relevants'**

R. We have removed the second "relevant".

**• l. 17: The last 1000-2000 years are normally named the 'Common Era' and not the 'preindustrial period'. 'Preindustrial' could lead to confusion with simulations using fixed preindustrial boundary conditions. Focusing here on the Common Era as a 'must' seems a bit arbitrary since also other periods would be of interest.**

R. We have modified the sentence to: *The index-timeseries must not only cover the historical period of the past 150 years but also the period of interest, e.g., the past 1000 to 2000 years (Common Era).*

**• l. 58: Which non-linear methods have shown promise? Is non-linearity really the main advantage here or could other factors also be important?**

R. E.g., data assimilation, some neural network configuration and random forest have been successfully applied for climate index and climate field reconstructions. A non-linear nature is likely only of advantage if the underlying problem is also non-linear. The non-parametric nature of the GP has the advantage that we do not need to make any assumptions about the underlying (non-)linearity. In the rewritten introduction we now devote more space to motivate our choice of the GPR and especially the embedding space.

**• l. 103-105: Which climate variable do you use to construct the pseudo-proxies (e.g. SST, surface temperature, near-surface air temperature)?**

R. 2m air temperature over land and SST over ocean. We have made this more explicit in the text:

*"Over land, we use 2m annual mean  air temperature, over ocean we use annual mean sea surface temperature. "*

• **l. 109-111: How do the SNR and the construction of pseudo-proxies compare to other pseudo-proxy studies?**

The amount of local interannual noise in real proxies is usually assessed by the correlation between the proxy time series and the instrumental time series. These correlations may be in the range 0.3 to 0.7. The amount of local noise used in other pseudo-proxy studies is within this range, as ours. We will include a sentence in the revised version:

*"This is a reasonable choice, as the correlation for real proxies with observations ranges from 0.3 to 0.7. The amount of white noise apllied here is also well within the range of other pseudoproxy studies(e.g., Smerdon 2012)"*

• **Figure 1a: From the color scale in (a), it is very difficult to distinguish the correlation of the different records. Maybe you can improve the color scale.**

We have chosen a different colour map and decreased the range of the colour bar to match the maximum and minimum correlation of the selected records with the AMVI. See updated Figures 1 and B1 (previously C1).

• **Figure 1c-e: Over which period are the cross-correlation, STD radio, and embedding distance computed?**

For the pseudoproxy records the entire 2000 years of the simulation, for all pairs including the AMVI only the most recent 150 years. We have included this information in the method section (new section 2.3.3):

*"The final distance matrix $D$ is then obtained by evaluating Eq.5 for all pairs of records. For all pairs of pseudoproxies, the distance is estimated from the entire simulation length. For calculating the distance between the AMVI and the pseudoproxies, we use only the last 150 years (years 1850 to 2000) and detrend both AMVI and pseudoproxies before the calculation. "*

• **Figure 1d: The 100 could be removed.**

We have removed it.

• **l. 132: Are Matern functions really 'very complex' kernels?**

We have changed it to '*more complex kernels*'

• **l. 139-140: The inference strategy is described very briefly here. Some more explanation could be useful for readers not that familiar with Bayesian inference**

We refer the reviewer to Appendix A, where we describe in more detail the calculation of the posterior predictive distribution.

• **l. 160: Is there a reason why the abbreviation SVGP is not adopted in the manuscrit and 'sparse GP' is used instead?**

This was done because we later only use "full and sparse *em*GP" when we refer to the GP in embedding space. In section 2.2.1 (now 2.3.5) we will stick instead to SVGP. In the remainder of the manuscript we will stick to the full and sparse emGP nomenclature.

**• l. 163: What is the 'Adam Optimiser'?**

R. The Adam optimiser is an algorithm to search the best model parameters according to a prescribed cost function. It belongs to the family of stochastic gradient descent algorithms and it is widely used in machine learning applications due to its favourable properties. Essentially, it takes into account the mean and standard deviations of the gradient of the cost function in previous optimization iterations to propose a new value of the parameters. We have added a sentence to text:

*"The hyperparameters are learned through optimisation with the Adam Optimiser, which is a stochastic gradient descent algorithm widely used in machine learning applications (Kingma 2014). We use the algorithm as provided by GPflow."*

**• l. 171: How is the AMVI formulated in proxy space since it is defined over a different spatial scale than the individual proxies and how does GPR-based climate index reconstruction work when the GP is formulated as function of the proxy values (=temperature?)?**

R. We interpret 'spatial scale' in this comment as 'amplitude of variability' In this particular application, the pseudo-proxies and the AMV index are all defined as near-surface air temperature, so that the range of variability is roughly similar for all of them. The reviewer is right that this condition cannot be generalised and for other reconstructions the proxies and target time series would need to be standardised to unit variance. We have included a brief explanation in the discussion.

*"In principle, the framework presented here can be applied to any climate index that exhibits significant correlations with local proxy sites. It is thus not limited to the AMVI application presented here. With real proxies, that do not all come in units of °C, it might make more sense to standardise all records to unit variance. "*

We have also added a more explicit paragraph about the GPR in proxy space, together with the new Figure 2 for visualisation:

*"As described for the PCR, classical climate-index reconstruction methods formulate their underlying statistical model in a way that the climate index is assumed to be a function of temperature, the proxy values or e.g. principal components thereof. In other words, the regression is performed in temperature/proxy/PC space; the proxies/PCs are the predictors and the climate index is the predictand. If we reconstruct the AMVI with GPR in this classical setup, the target AMVI becomes the posterior mean function and the covariance is estimated across the proxy space. With the trained GP model, the AMVI can be reconstructed by evaluating the GP at the proxy values that occurred during the reconstruction period. Fig.1a shows a schematic for the regression in proxy space for an example where the AMVI is given as a function of two pseudoproxy records $p\_1$ and $p\_2$. In this example, the posterior mean AMVI-function forms a surface in the space spanned by $p\_1$ and $p\_2$. Note that in our pseudoproxy experiments we use 23 pseudoproxies (Fig.1a), so the proxy space is actually 23-dimensional, which is impossible to visualise."*

**.• l. 177-192: I struggled to understand why the embedding space needs to have the given dimension and the idea of how the embedding space is constructed did not become clear to me until I finished reading Sect. 2.2.3.**

R. Indeed the new set-up of the GPR is not easy to visualise. We have rearranged and rewritten the entire previous section 2.2 (now section 2.3) and hope the derivation, design and motivation for the embedding space become clearer now. Regarding dimensionality, we have added a new figure (new Fig.2) and the following explanation:

*"To adequately reflect the distances between the proxy records and the AMVI, the embedding space needs to have a dimension of (q-1), where q is the number of proxies including the AMVI timeseries. This is easiest to understand if you imagine a case where all timeseries should have the same distance from each other. To arrange, e.g. three timseries with equal distances from each other, you need a two-dimensional space (spanned by x1and x2 in Fig.2b). [...]"*

• l. 203-205: **What are the properties that need to be fulfilled by the distance matrix / distance measure? Why is a positive correlation between records needed between the records?**

R. We have added the three criteria that need to be met:

*"To be used as a distance metric in MDS, a metric must meet the following three criteria: (1) it needs to be positive, (2) it needs to be zero, when it is applied on the object with itself and (3) it needs to be symmetric (e.g., Mead 1992)."*

Our chosen distance measure meets all three criteria. The CC needs to be positive because otherwise, two records with high negative CC would be placed further away even though they co-vary closely in time. Positive correlation could be easily ensured by changing the sign of one of the records, But all our selected records have only positive CCs anyway (Fig.1c).

• l. 209: **I guess that equidistant coordinates perform worse because than all records influence the AMVI roughly equally whereas for CC-based coordinates records with a high correlation with the AMVI become more important. But why does including the SR improve the result compared to just using CC?**

R. The reviewer is correct about the case with the equidistant coordinates, When performing the reconstructions only with CC-based coordinates, we found that the reconstruction is dominated by northern hemisphere records which have a much larger range of variability. This became especially important for networks with realistic proxy availability. Further back in time, only NH records are available and the reconstructed variability for earlier periods was strongly overestimated in the CC-based case. The consideration of the SR in the metric ensures that records that may have larger variability are considered farther away from the target, this improved the magnitude of the reconstructed variability. An alternative approach would be the normalisation of all records. In that case, CC-based coordinates would be sufficient. We have added a short explanation to the text:

*"With equidistant coordinates all records determine the AMVI to the same degree, regardless of their actual correlation with the AMVI. With a metric based solely on CC, the reconstruction is dominated by records with high variability and the resulting AMVI variability is overestimated. The additional SR-scaling yields improved variability estimates."*

• **Equation (3): Is RA = SR?**

R Yes we have corrected this

• l. 213-218: **The description was a bit short here for me to really understand what is happening and why.**

R. The rescaling of the time axis is done to ensure that the variations along the time axis are comparable to the variations along the other embedding dimensions. This is a necessary step due to how the kernel k2 is formulated, distance in time and embedding space must be comparable, to allow for interactions across time and records. Without the rescaling, either the distance between records or the timescale would dominate the lengthscale of k2. If the kernel would be formulated such that time and embedding dimensions were treated separately, this rescaling would likely not be

necessary. But as stated in the manuscript, the current kernel formulation outperforms the separated kernel. We have added this explanation to the text:

*"Because k2 operates on both the time and the embedding dimensions, we rescale the time steps to be of the same order of magnitude as the distances between the records. This is necessary to allow for the interaction across records and time. Otherwise, the lengthscale of k2 would either be dominated by the time step or by the embedding distance. One rescaled time step equals the mean of the distance matrix **D**."*

• **Equations (4/5): The Gaussian kernel functions lead to very smooth (infinitely differentiable) functions, likely much smoother than most processes actually observed in climatology. Does this lead to overly smooth predictions on certain timescales and did you test the procedure with kernels that lead to less smooth posteriors? It also might be useful to write down the final covariance model as an equation.**

R. The temporal component of Gaussian kernel leads to predictions that are indeed smooth in time, as it acts as a low-pass filter on the predictor time series. However, this does not mean that the model is not able to represent rapid changes of the target variable if the proxy records also change rapidly. The case that the reviewer raising - a non differentiable temporal behaviour - is in our opinion extremely rare. And as we have stated in the text, we use this very simple kernel formulation because we have no prior information that would justify more complex kernels. More complex kernels with less smooth posteriors would introduce more uncertainty and reduce interpretability.

The final covariance model / Kernel equation is simply the sum of k1 and k2. We have added this as an additional equation (new Eq. 8) in the manuscript

• **l. 230: Is σn 2 the same as a nugget effect in statistical modeling? If so, it might make sense to mention it here for the statistically-inclined readers.**

R. Yes, we thank the review for this suggestion. We have included this in the text:

*"Introducing sigma_n^2 is similar to the so-called nugget effect in geostatistics."*

• **l. 235/236: Do you have an explanation for why this slightly unusual formulation (two kernels acting in time but only one kernel acts in the "embedding space/distance") performs better than models without k1 or with separated kernels acting in time and embedding distance? Does this indicate that the system (AMVI) is better described by two characteristic timescales instead of one similar to two-box energy balance models outperforming one-box energy balance models in predicting sea surface temperatures?**

R The rationale is to allow for different time scales in the autocorrelation (k1) and in the cross-correlation between proxy records (k2). Also, the lengthscale of k2 cannot be interpreted as a pure timescale as it describes the typical lengthscale for the influence of the respective records across time and embedding space. We have added an explanation to the text:

*" The higher skill of this kernel makes sense, if you consider how the kernel design affects the interactions between the different timeseries. Having only k2 does not consider that the timescale of auto-correlation may not be the same as the timescale of cross-correlation. It therefore makes sense to have $k1$ operate across the time dimension only. If k2 operated only across the embedding dimensions, no interaction between different records across time would be possible."*

We think that this is not related with the issue raised by the reviewer, as our model does not include any type of forcing or reservoirs for the Atlantic Ocean. If we had included in the predictors a forcing

time series, then the interpretation of the time scale would indeed be closely related to a thermal inertial timescale of the ocean.

• **Equations (6) - (9) did not help my understanding. Either they should be embedded better in the text / explained better or they could be removed.**

R We have removed the equations. Instead we tried to explain the regression in proxy space and regression in embedding space more clearly and also included a figure to visualise the two spaces for a simplified example with three timeseries (two pseudoproxies and the AMVI). See new Fig. 2. And new sections 2.2 and 2.3 (rewritten version of old 2.2)

• **l. 259: There is a typo in 'obtain'**

R Noted and corrected

• **l. 299/300 (and similar parts in subsequent sections): The difference between the MPI- ESM- and CCSM4-based results could be explored a bit further. What are the main differences between the simulations that might explain the differing behavior of the reconstruction methods?**

R. See point number 1 in response to Major Comment 6.

• **l. 313/314: Is there an explanation for why the sparse GP performs better for noisy than for perfect pseudo-proxies on multi-decadal timescales?**

R. See point number 3 in response to Major Comment 6.

• **Fig. 4: This is an interesting figure to explain some of the differences between methods displayed in Fig. 2/3 but unless I missed something, it is barely discussed in the text.**

R. We have included more discussion of Fig.5 (previous Fig.4) in Section 3 and 4. (see tracked changes)

• **l. 359: Could the overestimation of variability in periods with few available proxy records be explained by relying too strongly on a small number of proxy records which tend to be more variable than the AMVI due to integrating over a smaller spatial scale? This could maybe be tested by comparing hyperparameters fitted separately for periods with high and low record availability.**

R. This is certainly a reasonable explanation. In fact, dendroclimatologists apply a statistical tool specially designed for this purpose called 'variance stabilization'. Our objective is, however, to compare the methods in different situations , i.e. is method A or B closer to the truth, and not so much the correction of the effect of sparser proxy networks. That would be a different methodological paper.

In our specific case, however, we think that the suggestion of the reviewer can only be part of the explanation. On the one hand we try to amend for the effect of few records with large variability through the additional SR scaling in the distance matrix. On the other hand, if this were the only explanation, we would expect the effect to get stronger the further we go back in time as the number of records decreases further. See also answer to comment on l.209 about CC-based coordinates and SR-scaling.

We have additionally performed a sensitivity experiment as suggested by the reviewer, and also explored the (non-)stationarity of the embedding distances. See points numbers 2 and 5 in response to Major Comment 6.

• l. 393-398: Is the introduction of a 'noise variance' parameter similar to error-in-variables approaches for frequentist regression models?

R. Formally, the noise variance is not the same as the error-in-variables approach. The EIV accounts for uncertainty in the independent variables/predictors, while the noise variance describes the uncertainty of the dependent variables/predictands. There are studies that explore how to incorporate measurement errors/input uncertainty in GPR (e.g. Zhou et al 2023, https://jmlr.org/papers/v24/21-1480.html), but the classical GPR assumes independent variables without error.

Conceptually one could argue that in our application, the noise variance is somewhat similar to the EIV approach, in the sense that the noise variance accounts for the uncertainty in both the pseudoproxies and the AMVI. And even though we have formulated the regression such that both pseudoproxies and the AMVI are the regression targets, the AMVI is still determined by the surrounding pseudoproxies in the embedding space. The EIV model, however, requires the knowledge of the ratio of noise in the independent and dependent variables, whereas here the noise variance is estimated along with the other hyperparameters,

• l. 408: Is the model really using one tenth of the available training data if you use every tenth time step but also an (optimized) subset of the original locations?

During the entire optimisation/fitting procedure, the model always uses a slightly different subset in every optimisation step, so in a sense it uses more than a tenth of the data. But the resulting co-variance matrix is based on an optimised subset which always corresponds to a tenth of the data.

• l. 416-418: Can you expand on these length scales and magnitudes? What are expected values and where do the parameters rank in the range of reasonable values?

R. We have expanded on the magnitudes:

"*The hyperparameters that have a straightforward physical interpretation, i.e. the typical lengthscale $l_{f,t}$ and variance $\sigma_{f,t}^2$ of the first kernel and the noise variance $\sigma_{n2}$, also appear reasonable in their magnitudes in most cases (red stars in Fig.5). The timescale of the full emGP is on the order of 2.7 years, which is a reasonable timescale of auto-correlation. As discussed above, $igma_{n2}$ captures the magnitude of the mean added noise variance across all records. The signal variance $\sigma_{f,t}^2=0.12$ indicates a temporal temperature variability of approximately 0.35 K. For all selected pseudoproxies, the temporal variability ranges from 0.28 to 1.15 K. The estimated $\sigma_{f,t}$ is thus on the lower end of plausible values. The timescale $l_{f,r}$ and variance $\sigma_{f,r}^2$ of the second kernel are less straightforward to interpret, as they operate across space and time. However, $\sigma_{f,r}^2$ should somehow reflect the mean temperature variability across all records and time. The estimate of $\sigma_{f,r}^2=0,51$ indicates a variability of 0.71. This is a fairly close estimate of the actual 0.82 K of the underlying data.*"

• l. 444: The GPR-model seems to handle white noise proxies very well, in parts due to the inclusion of the parameter $\sigma_n$. What would happen if the proxy noise would be auto-correlated? Is there a way to adapt the model accordingly?

R. The GPR model could be modified to account for temporal autocorrelation in $\sigma_n$. Actually, it seems that the GPR model would need to be cast in an embedding scape similar to the one implemented here (explicitly including the temporal dimension). This augmented model could be even expanded to account for spatial and temporal correlation of the $\sigma_n$ - in that case $\sigma_n$ would not be just a random variable but a gaussian random field itself, described by three additional

hyperparameters (the local noise, one temporal decorrelation and one spatial decorrelation). The calibration would certainly become more complex.

*"Here, we have only tested white-noise pseudoproxies, i.e. we assume that the noise in the pseudoproxy records is not correlated in time. The typical noise model for $\sigma_n^2$, which we apply here, also works with the assumption of uncorrelated Gaussian white noise. For real proxies this may not always be the case. There are ways of adapting the noise model to include, e.g., correlated noise (see Rasmussen & Williams, 2006). The embedding space would be a good starting point for this, as we explicitly take the time dimension into account. The noise model would introduce additional hyperparameters and make the calibration more complex. If we simply used our current set-up with correlated noise, the model might interpret some of the noise correlation incorrectly as actual data-correlation. This could be the subject of follow-up studies."*

• **Conclusions: Since this paper develops and tests a new methods for climate index reconstructions, it would be very useful for the reader to get some more guidelines for future applications of the method and how it might be applied to other indices.**

**R.** We have added a section to the discussion: "Using real proxies and wider applications". There we have added two new and modiefied paragraphs:

*"In principle, the framework presented here can be applied to any climate index that exhibits significant correlations with local proxy sites. It is thus not limited to the AMVI application presented here. With real proxies, that do not all come in units of °C (e.g., lake sediments, tree ring width, isotope ratios), it might make more sense to standardise all records to unit variance. In this case, the embedding distance would no longer need to include the SR-scaling. A simple dependence on the CC might be sufficient. This remains to be tested.*

*In order to use this framework for indexes that operate on longer timescales, it might become necessary to include records with lower temporal resolution. This would require subsampling of all records to the lowest common resolution, which is common practice in long-term reconstructions. It might also be possible to train one emGP model for the high-resolution records and one for the low resolution-records. The caveat here is that the observational period is often too short to include enough training data for the low-resolution records. But this is true for all reconstruction methods and not unique to the emGP framework here.*

*In the TCnpp cases, we created 30 different white noise realisations to estimate the noise-related uncertainty. With real proxies, we of course only have one realisation of the data and cannot run noise ensembles. But one could think of other ways of generating ensembles, e.g. with slightly different hyperparameters, slightly different ways of constructing the distance matrix or inclusion of different noise models for sigma_n^2. This would instead give insight into the other more methodological sources of uncertainty."*

**4. Review Figures:**

[Figure]

*Figure R1.1: Patterns of the leading EOFs derived from the simulated temperature in the North Atlantic-European sector by the models MSI-ESM and CCSM4. The percentage of explained variance is 27% and 41%, respectively. Note the different scalings in both panels - the leading EOF of the CCSM4 model has larger total variance and explains a higher percentage of the total variance than for the MPI-ESM model*

[Figure]

*Figure R1.2. The upper panel shows the newly fitted full emGP based on the period 1000-1500. The lower panel shows the original reconstruction from Figure 5 in the original manuscript version, where the hyperparameters were estimated from the period 1500-2000 (only the period 1000-1500 is shown here).*
This figure was added in the appendix (Fig.D2)

[Figure]

*Figure R1.3: Hyperparameters for the sensitivity experiments, analogously to Figure 4 from the original manuscript. Red diamonds correspond to the full emGPR fit based on the period 1000-1500 (upper panel in Fig.R1.2), grey diamonds to the second TCp2k case with perfect proxies and realistic proxy availability (trained on 1500-2000). Yellow stars correspond to the sparse emGP reconstruction with a network that contains only 11 pseudoproxies. Grey stars are from the analogous sparse emGP TCppp case from the manuscript. Yellow dots correspond to the same 11-proxy network but with added white noise. Grey dots are the hyperparameters from the corresponding original TCnpp sparse emGP fit, for comparison.*

A modified version of figure was added in the appendix (Fig.D3)

[Figure]

*Figure R1.4: Sensitivity experiments with 50% smaller networks. Left panels with MPIESM-based pseudoproxies, right panels with CCSm4-based pseudoproxies. Upper panels with perfect pseudoproxies, lower panels with noisy pseudoproxies. The hyperparameters for the MPIESM-based experiments can be found in Fig.R1.3 (yellow stars and dots). For the CCSM4-based experiments, the hyperparameters can be found in Fig.R1.5 (yellow stars and dots).*

This figure was added in the appendix (Fig.D1)

[Figure]

*Figure R1.5: Hyperparameters for the CCSM4-based TCppp (stars) and TCnpp (dots) experiments with twelve randomly selected pseudoproxies (yellow) and the original 24 pseudoproxies (grey).*

[Figure]

*FIgure R1.6: Changes in the embedding distance metric Dij for all pseudoproxies and the AMVI over time (x-axis in years). The distances are calculated with a running window of 151 years. The anomalies are calculated against the mean distance over the entire period- Red indicates greater distance (less similarity), blue indicates smaller distance (more similar). The number of records (y-axis) corresponds to the numbers in Fig. 1 of the manuscript.*

This figure was added in the appendix (Fig.D4)

**Reviewer #2**

We would like to thank Referee 2 for their detailed and constructive comments. In the following to explain how we plan to revise the manuscript to address their suggestions.

**The original comments are written in bold font**, our responses with normal font and in colour. Modified text passages are highlighted with *italics.*

**C The quality of the paper is generally high, the analysis appears accurate from my knowledge, and there are no major points which I think should prevent its publication. There are however several minor points and suggestions which I identified and reported below, which I think could improve the paper further.**

R. We thank the reviewer for the general positive assessment

**Lines 33-34: "however" is awkward in the middle of the negation.**

The sentence has been removed during revisions in response to Reviewer 1

**C Line 134: Might be useful to describe the term 'hyperparameter' for those outside the machine learning field, how is it different from a normal 'parameter'?**

R. The reviewer is right. This notation is usual in the Gaussian Process literature, but can be confusing. A "normal" parameter would directly be related to describe the underlying target function, such as e.g. a regression coefficient in a linear least squares model. The GP-model itself however is non-parametric. To make this distinction, kernel parameters are therefore often referred to as hyperparameters. We have added the following sentence:

*[...] the specific form is determined by the kernel parameters. Since the underlying GP model itself is non-parametric, kernel parameters are typically referred to as hyperparameters (Rasmussen & Williams, 2006).*

**C.Lines 148: So the batch size is the total number of observations across records (i.e. 5 records with 100 observations plus 2 records with 200 observations would mean a batch size of 900)?**

R. The batch size corresponds to the number of training observations given to the algorithm in. In our case it is defined as suggested by the reviewer.

**C. Lines 180-183: How are the irregular resolution of the proxies handled (for the realistic P2k case)? Or are the pseudo-proxies created at annual resolution? In which case it would be helpful to briefly hint in the discussion how realistic irregular proxies could be used in the future and how it might affect the results.**

In the p2k case, the pseudoproxies are indeed created with annual resolution. We have added a corresponding sentence to Section 2.1:

*"In all three testcases, the pseudoproxy records have annual resolution"*

In fact, at all 23 selected locations, the corresponding real-world proxies also have annual resolution or higher. We have therefore removed the 5year-criterion from the text, as it was creating confusion.

We have also included the handling of records with different resolution in the discussion.

**C. Equation 3: Should RA be SR?**

Yes, we have corrected it.

**C. Lines 248-249: So the embedding coordinates are calculated from the distance matrix via multidimensional scaling? Might not be obvious for the layman (including me) how the matrix is obtained, could it be made more explicit in an Appendix?**

We have rewritten large parts of the method section and hope that the definition of the distance matrix is clearer now. The distance matrix is obtained be evaluating Eq.4 (previously Eq.2) for all pairs of proxyrecords and proxyrecords with the AMV. There is no unique distance metric to define the distance matrix. We use the one described in Eq.4, which seems to be a reasonable choice. But other metrics could be possible, too.

We have also included a sentence and a reference on multidimensional scaling.

**C Line 259: "obatain"**

R We have corrected the typo.

**C Equation 6,7,8,9: Shouldn't the matrix be equal to something for an equation? Make explicit which one is the input and which the output.**

R .We have removed the equations and instead tried to explain the respective regression spaces and also added a visualisation of both regression spaces for a simplified example with three timeseries (two pseudoproxies and the AMVI).

**C Lines 299-300: Could it be because CCSM4 is more homogeneous (less spatial degrees of freedom) since the mean embedding distance between the records is smaller? Just a thought.**

R. Yes, the reviewer is correct. The first EOF of the CCSM4 surface temperature explains a much larger portion of the variance than in the MPIESM case (see also Figure R1.1. in response to Reviewer 1). This could explain the better performance of the PCR in the TCppp CCSM4 case. It also indicates that the emGPR seems to perform equally well regardless of the spatial coherence of the underlying temperature fields.

We have added a short paragraph on this:

" *The superior performance of the PCR in the CCSM4 case can be explained by a greater spatial coherence of the underlying CCSM4 temperature field. The leading EOF explains 41\% of the total variance in the CCSM4 case and only 27\% in the MPIESM case (not shown). The difference in spatial coherence is also reflected in the overall smaller embedding distances in the CCSM4 case (compare Fig.1 and B1). The fact that the full and sparse emGPR perform about equally well for MPIESM and CCSM4, indicates that the emGPR is more robust to different degrees of spatial coherence in the underlying field. This can be considered an additional strength of the emGPR. *"

**C Figure 2: Unclear to me what the 95% confidence intervals represent in the temporal domain? Are they the spread of the unsmoothed data? Also for the spectra it is not explained what the confidence intervals are, simply chi-square CI with 2 degrees of freedom? It is a question of style and not necessary, but I personally like smoothing the spectra to make for a clearer comparison (e.g. Using a Gaussian smoothing kernel as in JW Kirchner, Aliasing in 1/f(α) noise spectra: Origins, consequences, and remedies. Phys Rev E Stat Nonlin Soft Matter Phys 71, 066110; 2005).**

R  In case of the full and sparse emGPR, the shading indicates the 2sigma range of the posterior GP distribution. This is the uncertainty range directly estimated by the GP. For the PCR, the CI corresponds to the uncertainty in the estimated regression coefficients, which is based on the t-distribution. The spectral uncertainty does indeed correspond to the chi-square CI. We have added this information to the caption:

*"The CI is determined by the posterior GP distribution for the full and sparse emGP. For the PCR, the CI is derived from the uncertainty in the regression coefficients, which is based on the t-distribution. [...] Shading indicates the 95\% confidence interval as obtained from the Chi^2-distribution."*

We have also added some more discussion of the uncertainty ranges , as requested by reviewer 1.

**C. Caption Figure 2 and 3: "powerspectra" -> "power spectra"**

R Noted and corrected

**C Figure 3: I don't understand the indicated 95% CI number. Do those correspond to the same CI shown on Figure 2?**

R Yes, for each individual ensemble member, the CI is defined the same way as in Fig.2. The values reported by the respective numbers correspond to the CI averaged over time and all ensemble members. We have added this information to the caption:

*"The 95% CI in the lower right of all three panels indicates the CI averaged over time and all ensemble members."*

**C Section 3.2: I would generally favour calculating the statistics for individual ensemble members and reporting the mean+/- standard deviation rather than calculating them with respect to the ensemble mean. Similarly, I would show the mean of the spectra rather than the spectrum of the mean for Figure 3 b,d,f as it is more representative of the real result one would obtain.**

Point taken. We have removed the ensemble mean reconstructions (as well as the spectrum of the mean) from the figures, and instead of the metrics of the ensemble mean, we present the mean and spread of the metrics from the individual ensemble members. This information was already partly available in the first versions of the manuscript (old Fig. B1 for the MPIESM TCnpp case), but we have now calculated it for all noisy reconstructions and modified all figures and text accordingly.  Some of the numbers have slightly changed, but the conclusions have remained the same.

Appendix C now contains the distribution of the skill metrics for all noisy reconstruction ensembles (MPIESM TCnpp, MPIESM TCp2k and CSESM TCnpp).

Text was adjusted mostly in Section 3.2 and the relevant paragraphs of 3.3.1 and 3.3.2.

**C Figure 3 b,d,f: I wonder whether the PCR has the right high-frequency amplitude for the right reason? Are the high-frequencies just noise and thus PCR doesn't perform better than the other methods or are they actually correlated with the real series?**

R This is a behaviour that has been found in other previous analyses. Yes, the PCR reconstructions are indeed correlated with the target at high frequencies.

**C Lines 329-331: Do the authors have an idea why the sparse can outperform the full GP? Could it be a case of overfitting when noise is present? Such that the sparse one is less sensitive to overfitting?**

See our points 3 and 4 in our response to major comment 6 by reviewer #1. We have included corresponding paragraphs in the discussion.

**C Lines 335-339: How are data resolution handled? If there is 5 years resolution, then are there gaps between the years or are the values interpolated? Or are the annual data used and only clipped at the end of the record?**

R The latter case. In the manuscript only proxies with annual resolution are created and clipped at the end of the record.. This is the standard case for past millennium reconstructions. We have added a sentence to the text:

*"To achieve realistic data availability, we clip the annually resolved pseudoproxy records at the start and end years of the corresponding real-world proxy records from the PAGES2k data-base."*

As stated above, we have also removed the 5-year resolution selection-criterion, because at all 23 selected locations, the real proxy records have annual resolution or higher.

**C Line 387: I would remind the reader for the discussion what AMV-relevant timescales are. Maybe write in parenthesis something like (decadal to multi-decadal).**

R. We have added the suggested parentheses.

**C One issue I would like better discussed is the loss of variance on longer than centennial timescales in the sparse emGPR for the full 2k run. To me this is quite an important limitation since I don't think it makes sense to restrict AMV-relevant timescales to decadal to multi-decadal; there is a continuum of processes and I don't think there are reasons to believe that it would flatten out on longer than centennial timescales or be related to a separate non-AMV relevant process right?**

R. The length of the available simulations is 1000-2000 years, so that it is difficult to assess the behavior on multicentennial timescales - the degrees of freedom is considerably reduced. For instance, the correlation between reconstruction and target woud be estimated using, say, 10 degrees of freedom, assuming that centuries are independent samples.

Note however, that in case of the full emGP, the underestimation on very long timescales was an artefact of taking the spectrum of the ensemble mean. In the new versions of the figures, where we have removed the spectrum of the mean, you can see that actually all ensemble members (at least of the full emGP) capture the longterm variability well (new Fig.8c). We have therefore updated the text in section 3.3.1 accordingly.

**C Line 465-466: Looking forward to seeing how it compares to the traditional PCR method!**

R. Thank you. We are also curious to see how it will compare to existing reconstructions. This is ongoing work and will be the focus of a follow up study.

---

## Author Response (AR2)

We thank the editor and the two anonymous reviewers for the positive evaluation.

We have corrected the manuscript according to the list of Reviewer 1. The corrections and a small number of additional editorial changes can be found in the manuscript with tracked changes.